# *Lactobacillus rhamnosus* colonisation antagonizes *Candida albicans* by forcing metabolic adaptations that compromise pathogenicity

Raquel Alonso-Roman[1], Antonia Last[1], Mohammad H. Mirhakkak[2], Jakob L. Sprague[1], Lars Möller[1], Peter Großmann [2], Katja Graf[1,3], Rena Gratz[1], Selene Mogavero [1], Slavena Vylkova[4], Gianni Panagiotou[2,5], Sascha Schäuble[2], Bernhard Hube [1,6 ✉] & Mark S. Gresnigt [7]

Intestinal microbiota dysbiosis can initiate overgrowth of commensal *Candida* species – a major predisposing factor for disseminated candidiasis. Commensal bacteria such as *Lactobacillus rhamnosus* can antagonize *Candida albicans* pathogenicity. Here, we investigate the interplay between *C. albicans*, *L. rhamnosus*, and intestinal epithelial cells by integrating transcriptional and metabolic profiling, and reverse genetics. Untargeted metabolomics and in silico modelling indicate that intestinal epithelial cells foster bacterial growth metabolically, leading to bacterial production of antivirulence compounds. In addition, bacterial growth modifies the metabolic environment, including removal of *C. albicans*' favoured nutrient sources. This is accompanied by transcriptional and metabolic changes in *C. albicans*, including altered expression of virulence-related genes. Our results indicate that intestinal colonization with bacteria can antagonize *C. albicans* by reshaping the metabolic environment, forcing metabolic adaptations that reduce fungal pathogenicity.

[1] Department of Microbial Pathogenicity Mechanisms, Leibniz Institute for Natural Product Research and Infection Biology - Hans-Knoell-Institute, Jena, Germany. [2] Systems Biology and Bioinformatics Unit, Leibniz Institute for Natural Product Research and Infection Biology - Hans-Knoell-Institute, Jena, Germany. [3] Dynamic42 GmbH, Jena, Germany. [4] Septomics Research Centre, Leibniz Institute for Natural Product Research and Infection Biology – Hans-Knoell-Institute, Jena, Germany. [5] Department of Medicine and State Key Laboratory of Pharmaceutical Biotechnology, University of Hong Kong, Hong Kong, China. [6] Institute of Microbiology, Friedrich Schiller University, Jena, Germany. [7] Junior Research Group Adaptive Pathogenicity Strategies, Leibniz Institute for Natural Product Research and Infection Biology - Hans-Knoell-Institute, Jena, Germany. ✉email: bernhard.hube@hki-jena.de

The yeast *Candida albicans* is a common commensal of the intestinal mycobiota[1,2]. However, certain predisposing conditions, such as a dysbalanced microbiota and a compromised immune system can favour its shift from a commensal to a pathogenic stage[3]. Consequently, *C. albicans* can translocate through the intestinal epithelium into the bloodstream[4,5], resulting in disseminated candidiasis associated with high mortality rates[6]. Specifically, an intact epithelial barrier, the immune system, and the commensal intestinal microbiota play crucial roles in maintaining *C. albicans* commensalism by suppressing overgrowth and pathogenicity[3,7,8]. Bacteria belonging to different genera, such as *Bifidobacterium*, *Streptococcus* and, more extensively, *Lactobacillus*, have been studied with regard to their antagonistic effects against *C. albicans*[9,10].

*Lactobacillus* species have been discussed as oral probiotics to prevent candidemia in premature newborns[11,12] and are also being explored as potential therapies against vulvovaginal candidiasis[13–15]. In vitro studies have demonstrated that lactobacilli can antagonize proliferation[9], hypha formation[16–21], biofilm formation[18,20–23], or even kill *C. albicans*[24,25]. However, these studies often do not recapitulate the role of the host or its epithelial barriers as active partners in the interaction. Previously, we demonstrated that the antagonistic effects of *L. rhamnosus* on intestinal epithelial cells (IECs) are associated with reduced *C. albicans* invasion, and damage[26]. *L. rhamnosus* colonization of IECs prevented fungal outgrowth and reduced the number of host cell-associated *C. albicans* cells through shedding[26]. Similarly, *L. rhamnosus* colonization in an intestine-on-chip-model reduced the fungal burden, epithelial damage, and fungal translocation into a surrogate bloodstream compartment[27].

However, the molecular mechanisms underlying the antagonistic effects of *L. rhamnosus* colonization against *C. albicans* pathogenicity remain, to a large extent, unresolved. Several antagonistic mechanisms of lactobacilli have been described using models excluding host cells, such as peroxide production[28] or lactic and acetic acid production[25,29]. However, these antagonistic mechanisms did not play a role in the protection of host cells[26]. More complex in vitro models are required to dissect the interactions between the host, lactobacilli, and *C. albicans* in a physiologically relevant manner to understand the principles of the observed protective activities.

In this work, we took a systems-biology approach involving multi-omics (transcriptome/metabolome) profiling, in silico metabolic modelling, and in vitro infection biology to uncover how *L. rhamnosus* colonization of IECs mediates protection against *C. albicans* infection. This included dissection of the metabolic crosstalk between epithelial cells, *L. rhamnosus*, and *C. albicans*. Using this approach, we elucidated how multifactorial changes in the metabolic environment force *C. albicans* transcriptional adaptation, resulting in a reduced capacity to damage epithelial cells.

## Results

### *L. rhamnosus* suppresses *C. albicans* pathogenicity.
We recently showed that *L. rhamnosus* reduced the number of *C. albicans* cells in contact with IECs[26], which may reduce invasion and damage of the tissue. Infections with a lower number of *C. albicans* cells also resulted in a reduced fungal burden in the tissue after infection (Fig. 1a). Interestingly, this reduced fungal burden in the absence of *L. rhamnosus* still induced more damage than when lactobacilli were present (Fig. 1b), demonstrating that reduced numbers of host-cell-associated *C. albicans* alone are insufficient to mediate protection.

We previously showed that killed *L. rhamnosus* failed to protect IECs against *C. albicans*-induced damage[26]. To confirm that colonization is only protective when lactobacilli are metabolically active and proliferating, the bacteria were killed using antibiotics 4 hours post infection (hpi). Killing *L. rhamnosus* after colonization still abolished their protective effect (Fig. 1c). However, physical separation of *L. rhamnosus* from IECs and *C. albicans* did not impair the protection (Fig. 1d), suggesting that the environment plays a key role.

### Host-bacterial-fungal metabolic crosstalk.
A detailed insight into the metabolic interactions between *C. albicans*, *L. rhamnosus*, and IECs was obtained using untargeted metabolomics. In the supernatants, 235 metabolites were identified, and their relative abundances were quantified. From these metabolites, 89 were present in the culture media, suggesting that the remaining metabolites are derived from metabolic activity of IECs, *C. albicans*, and *L. rhamnosus* (Supplementary Data 1).

The metabolic environment slightly differed between uninfected IECs and *C. albicans*-infected IECs (Fig. 2a I). However, at 6 and 12 hpi *L. rhamnosus* colonization of IECs removed 53 and 46 metabolites and resulted in the appearance of 73 and 74 metabolites, compared to uninfected IECs, respectively (Fig. 2a II, b). Interestingly, during *C. albicans* infection of *L. rhamnosus*-colonized IECs only 12 metabolites were altered. (Fig. 2a III, b). Unsupervised hierarchical clustering of the metabolome revealed distinct clusters of metabolites in the model at 6 hpi (Fig. 2c, Tab. S1) and 12 hpi (Fig. S1, Table S2), which are associated with specific dynamics. At 6 hpi, cluster 2 and 3 (Fig. 2c) represent metabolites produced by IECs. The metabolites in cluster 3, however, are no longer present following *L. rhamnosus* colonization, suggesting that they are consumed by *L. rhamnosus*. Metabolites in cluster 1 were only present when *L. rhamnosus* colonized IECs, and thus may be involved in their protective effect. Similar clusters were identified at 12 hpi (Fig. S1). Enrichment analysis showed that at both 6 and 12 hpi, cluster 1 was enriched in metabolites involved in pyrimidine metabolites. At 12 hpi, cluster 4 (cluster 3 at 6 hpi) was enriched in metabolites for valine, leucine, and isoleucine biosynthesis (Fig. S2).

### Intestinal epithelial cells foster *L. rhamnosus* growth.
*L. rhamnosus* cannot grow in Keratinocyte Basal Medium (KBM) unless IECs are present, yet active growth is essential for their antagonistic potential (Fig. 1c)[26]. Physical separation of *L. rhamnosus* from the epithelial cells did not compromise protection (Fig. 1d) nor bacterial growth (Fig. 2d). Likewise, IEC-spent medium supported *L. rhamnosus* growth (Fig. 2d), validating that epithelial-secreted metabolites foster bacterial growth. By simulating *L. rhamnosus* biomass formation in silico with flux balance analysis (FBA)[30], IEC supernatants were predicted to sustain an increased bacterial biomass, while the culture medium (KBM) did not (Fig. 2e, Supplementary Data 2). Thus, the IEC-derived metabolites, which disappear upon colonization (Fig. 2c cluster 3, Table S1), are likely those that foster *L. rhamnosus* growth. From the different metabolite types in this cluster at least one representative was tested in vitro for its capacity to foster *L. rhamnosus* growth. Supplementation with fructose, xanthine, nicotinamide, nicotinic acid, 3-methyl-2-oxobutyrate, 3-methyl-2-oxovalerate, N-acetylgalactosamine, carnosine, and amino acids did not promote growth individually (Fig. 2f). Carnitine and citric acid supplementation slightly supported *L. rhamnosus* growth (Fig. 2f), whereas combinations of citric acid with gamma-glutamylalanine or carnitine, and especially all three in combination, supported *L. rhamnosus* growth to rates similar as on IECs (Fig. 2f).

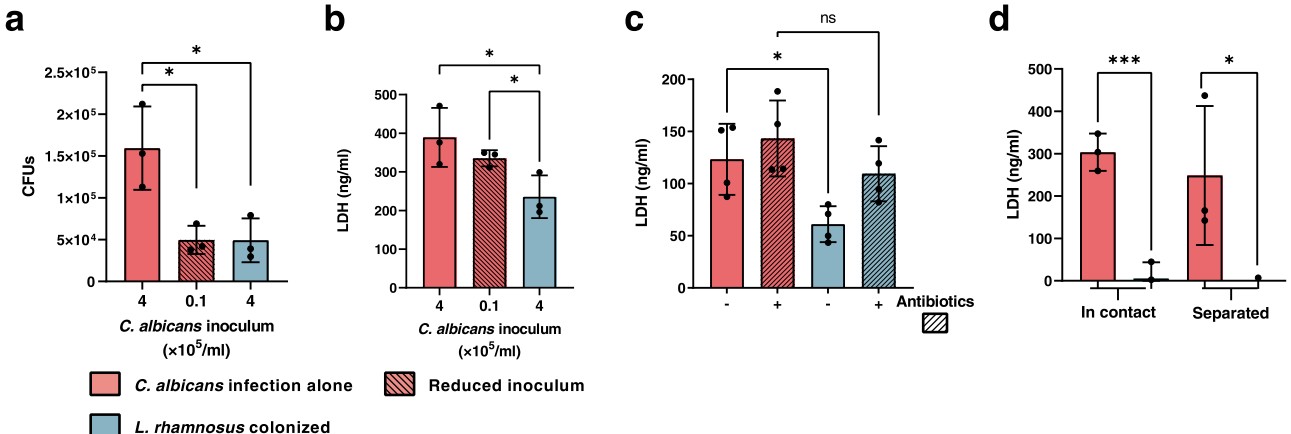

**Fig. 1 Reduced *C. albicans* inocula cause significant damage compared to *C. albicans* in the presence of live *L. rhamnosus* that reduces pathogenicity contact-independently. a** Fungal burden assessed by quantification of *C. albicans* CFUs and **b** the necrotic cell damage of IECs quantified by LDH activity in supernatants at 24 h post infection (hpi). Cells were infected with a *C. albicans* infection inoculum ($4 \times 10^5$/ml) in the presence and absence of *L. rhamnosus*, or with a reduced infection inoculum ($1 \times 10^4$/ml) in the absence of *L. rhamnosus* ($n = 3$ biological repeats; *L. rhamnosus* colonized A * = $p$ 0.0276, B * = $p$ 0.0434 reduced inoculum A * = $p$ 0.0225; B * = $p$ 0.0434). **c, d** Necrotic cell damage of IECs quantified by LDH activity in supernatants at 24 hpi, (**c**) in the presence and absence of *L. rhamnosus* colonization and antibiotic treatment with Gentamicin and Penicillin/Streptomycin at 4 hpi ($n = 4$ biological repeats; * = $p$ 0.0173 ns = 0.184), or (**d**) in the presence and absence of *L. rhamnosus* colonization where *L. rhamnosus* was in direct contact with the cells or physically separated using a transwell insert with a 0.4 μm pore size ($n = 3$ biological repeats; *** = $p$ 0.0009, * = 0.048). Bars represent the mean and standard deviation of the independent experiments, dots represent the mean of the technical replicates of the individual experiments, biological repeats were compared for significance using an unpaired t-test (two-tailed, one-sample), * = $p \leq 0.05$, ** = $p \leq 0.01$, *** = $p \leq 0.001$. Source data are provided as a Source Data file.

Additionally, reconstructed genome-scale metabolic models (GEMs) were simulated for IECs, *C. albicans*, or *L. rhamnosus*[31,32], monitoring both biomass production and feasible reaction fluxes by again applying FBA and also flux variability[33] analysis (FVA, Supplementary Data 2). Since IECs are persistently present in the infection model, GEMs were parametrized with supernatant-specific metabolites. We investigated the GEMs used for IECs, *C. albicans*, and *L. rhamnosus* to uncover reaction flux shifts in individual metabolic pathways across different nutritional conditions including blank media and supernatants of IECs, *L. rhamnosus* or *C. albicans* (Figs. 2g; S3). The in silico simulation using FVA revealed markedly altered *L. rhamnosus* metabolic pathway activities on IEC supernatants *vs.* culture medium, including nucleotide, Nicotinamide adenine dinucleotide (NAD), lipid-related metabolism, and Coenzyme A (CoA) synthesis. In addition, diverse amino-acid metabolic pathways showed differing feasible flux ranges in response to IEC-derived metabolite profiles (Fig. 2g). In comparison, changes in metabolic pathway activity were less prevalent in IECs (Fig. S3A). Here, only a few amino acids (including tyrosine and phenylalanine), ubiquinone, and taurine pathways next to the generic protein assembly/degradation metabolic subsystem showed shifts in pathway activity of at least 40% upon *L. rhamnosus* affected supernatants. These simulation results suggest that *L. rhamnosus* utilizes IEC-secreted metabolites without inducing drastic metabolic changes in the host. These results did not change upon varying thresholds for required flux activity changes in the compared conditions (Supplementary Data 3).

***L. rhamnosus*-derived metabolites antagonize *C. albicans*.** *L. rhamnosus*-mediated protection against *C. albicans* cytotoxicity is contact-independent (Fig. 1d), suggesting that soluble molecules mediate the antagonistic effects. To test this hypothesis, *L. rhamnosus* was grown independent of host cells in the supplemented KBM and *C. albicans* was grown in the conditioned supernatants. The *L. rhamnosus*-conditioned supernatants, reduced *C. albicans* filamentation and even induced a subtle

transition from hyphae to yeast (Fig. S4A). Metabolites specifically secreted when *L. rhamnosus* colonized IECs (cluster 1, Fig. 2c, cluster 1 Fig. S1B, Table S2) were investigated for their effects against *C. albicans*. Literature searches revealed that several of these metabolites, including phenyllactic acid[34], mevalonolactone[35], 2-hydroxyisocaproic acid (HICA)[36,37], and 3-hydroxyoctanoate[38], have previously been reported to have antifungal potential. We validated the effects of phenyllactic acid and 2-hydroxyisocaproic acid, and observed that these metabolites changed the pH of the medium. Nevertheless, compared to pH-adjusted medium, an impact on *C. albicans* filamentation was still observed. Further screening of the *L. rhamnosus* colonization-derived metabolites revealed several additional metabolites that differentially influenced *C. albicans* filamentous growth (Figs. S4B, 3a). Notably, cytosine induced a hypha-to-yeast transition, characterized by a complete halt of hyphal growth at approximately 8 h, after which cells continued to proliferate in the yeast morphology (Fig. 3a, Supplementary Videos 1 and 2). The metabolites identified also reduced filamentation and induced a subtle shift towards yeast growth when administered at lower concentrations and in combination with each other (Fig. 3b). After 4 h, hyphae were significantly shorter when the combination of metabolites was present (Fig. 3c). In line with the impaired filamentation, *C. albicans*-induced damage of IECs at 24 hpi was reduced (Fig. 3d). Causing the most striking phenotype, cytosine on its own was evaluated for its potential to inhibit *C. albicans*-induced damage and translocation. Both host-cell damage as well as *C. albicans* translocation were reduced (Fig. 3e, f).

***L. rhamnosus* induces a hostile environment for *C. albicans*.** The reduced *C. albicans* growth observed when infecting *L. rhamnosus*-colonized IECs[26] was also reflected by in silico FBA[30] of the genome-scale metabolic model of *C. albicans*. By using our established metabolome data-driven genome-scale metabolic models of IECs and *L. rhamnosus*[31,32] we predicted potential metabolite secretion or uptake (Supplementary Data 2 and 3).

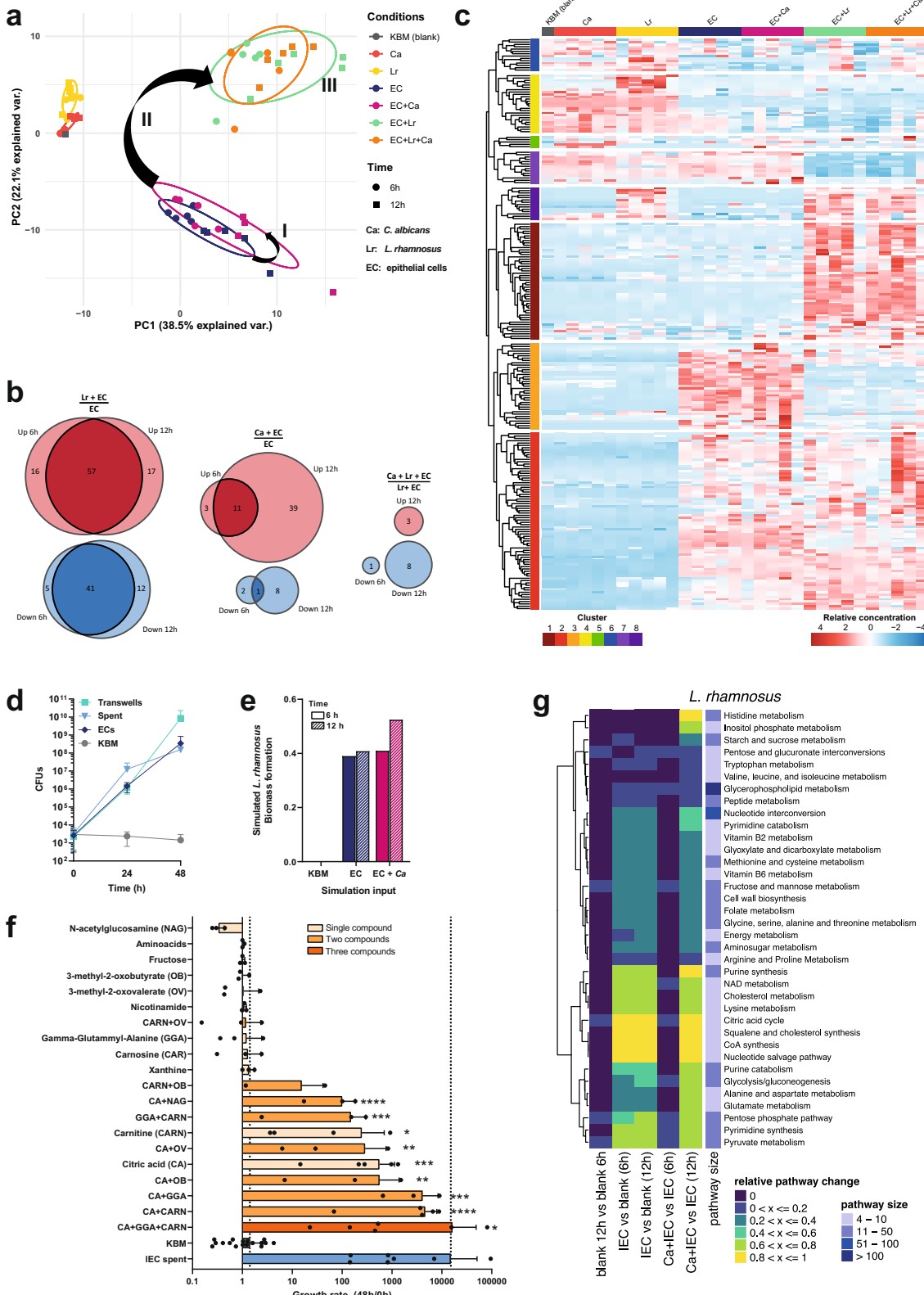

These predictions were cross-checked with phenotypic growth data of *C. albican*[39] and metabolomics fold changes for different conditions (Fig. S5).

Glucose, fructose, mannose, and N-acetylglucosamine/N-acetylgalactosamine support efficient *C. albicans* growth[39]. However, these metabolites were depleted upon *L. rhamnosus* colonization (Figs. 3g, S5). Instead, alternative carbon sources,

like 3-4-hydroxyphenyllactate, lactate, *N*-acetylglutamate, and malate showed approximately 5-fold higher concentrations, but these supported *C. albicans* growth to a lesser degree[39].

Amino acids are favourable nitrogen sources to sustain *C. albicans* growth[39] as well as carbon sources when glucose is consumed, yet *L. rhamnosus* colonization reduced the availability of most favoured amino acids (Fig. 3g) and increased

**Fig. 2 IECs metabolically foster *L. rhamnosus* growth. a** Principal component analysis of metabolite composition assessed by untargeted metabolomics in supernatants collected at 6 and 12 hpi. Arrows (I, II, III) indicate shifts in the metabolic profile. **b** Differentially increased or decreased metabolites in comparison of the different conditions shown in proportional Venn diagrams. Data summarized from *n* = 5 at 6 and 12 hpi. **c** Hierarchical clustering based on Euclidean distance of relative metabolite abundance in the supernatants at 6 hpi. **d** Growth of *L. rhamnosus* in KBM, on IECs, in transwells physically separated from IECs, or in supernatants of IECs (spent) assessed by counting CFUs on MRS agar. Data are shown as the mean and standard deviation (SD) of *n* = 3 biological replicates. **e** In silico prediction of *L. rhamnosus* biomass formation in KBM or supernatants of IECs. **f** Growth of *L. rhamnosus* assessed by CFUs after 48 h incubation in KBM supplemented with single metabolites or combinations of metabolites. Data of *n* = 3–6, KBM = 17, IEC spend *n* = 8 biological replicates are shown as the mean and SD. Data were tested for significance using a *t* test (two tailed, one-sample) against growth in KBM, * = *p* ≤ 0.05, ** = *p* ≤ 0.01, *** = *p* ≤ 0.001, **** = *p* ≤ 0.0001 (CA + GGA + CARN *n* = 6, * = *p* 0.0410; CA + CARN *n* = 5, **** = *p* < 0.0001; CA + GGA *n* = 3, *** = *p* 0.0003; CA + OB *n* = 3, ** = *p* 0.0041; CA *n* = 5, *** = *p* 0.0003; CA + OV *n* = 3, ** = *p* 0.0089, CARN *n* = 4, * = *p* 0.0218; GGA + CARN *n* = 3, *** = *p* 0.0001, CA + NAG *n* = 3, **** = *p* < 0.0001). Source data are provided as a Source Data file. **g** Comparison of metabolic pathway activity levels between different conditions as indicated. Relative pathway change was determined by identifying the number of pathway-specific reactions for which feasible flux ranges differ according to flux variability analysis.

N-acetylated amino acids. In silico GEM based analyses predicted that amino acids are taken up by IECs and *L. rhamnosus* in the majority of cases (Fig. S5). The relative decrease of amino acids upon *C. albicans* infection suggests that *C. albicans* uses them as an alternative, but less-favoured carbon source. This was accompanied by a depletion of the favoured phosphorus source choline phosphate during *L. rhamnosus* colonization (Fig. 3g).

Collectively, in silico and metabolomics analyses revealed that *L. rhamnosus* colonization depleted favoured carbon, nitrogen, and phosphorus sources of *C. albicans*, which were replaced by alternative carbon or nitrogen sources forcing *C. albicans* into a suboptimal growth milieu. This could explain the 3-fold growth reduction when *C. albicans* infects *L. rhamnosus*-colonized epithelium (Fig. 1a).

Further in silico GEM simulations provided clues on activity of individual pathways in both *C. albicans* and IECs (Fig. S3). Reaction flux ranges of *C. albicans* glycolysis, TCA cycle, or pyruvate metabolic pathways changed by 20–40% when simulated on colonized *vs.* uncolonized IECs supernatant (Fig. S3B). This suggests shifts in *C. albicans* metabolism due to the altered metabolic environment. *C. albicans* oxidative phosphorylation was predicted to substantially alter in IECs supernatants, and even further change in the presence of *L. rhamnosus*, adding evidence that its energy maintenance might be affected (Fig. S3B). Additional changes were predicted in lipid, sulfur, and nucleotide related metabolic pathways, including biotin, butanoate, purine, and pyrimidine metabolism, suggesting a comprehensive metabolic shift in *C. albicans*, which begins to utilize alternative carbon sources.

**L. rhamnosus forces fungal transcriptional metabolic adaptation.** In silico simulations suggested that *C. albicans* adapts to cope with the changed metabolic environment. We hypothesized that these metabolic adaptations require transcriptional reprogramming. Differential gene expression of *C. albicans* during infection of *L. rhamnosus*-colonized IECs was investigated by transcriptional profiling. Unsupervised hierarchical clustering (Fig. S6A) and principal component analysis (PCA) (Fig. 4a) revealed a distinct gene expression pattern upon *L. rhamnosus* colonization, but only 1.3% of the up-regulated and 1.6% of the down-regulated genes overlapped across time points (Fig. 4b, c).

*L. rhamnosus* killed with antibiotics at 4 hpi did not induce noticeable transcriptional reprogramming of *C. albicans*, as exemplified by missing shifts in the first two PCA-defining principal components and gene expression profiles, (Figs. 4d, S6B), underlining that only live *L. rhamnosus* enforces transcriptional adaptation.

Next, we explored the transcriptional changes for hints pointing towards fungal metabolic adaptation. Gene Ontology (GO) enrichment analysis revealed that a variety of metabolic processes were down-regulated at 6 hpi, while the opposite was observed at 24 hpi (Fig. 5a). As the GO terms did not yield very specific insights into the transcriptional metabolic adaptation, we analysed the expression of transcription factors regulating metabolic adaptation as well as key metabolic processes.

Consistent with the limited favoured carbohydrate availability (Figs. 3g, S5), the transcription factor genes *MIG1* and *TYE7* were significantly up-regulated at 6 hpi, possibly to compete for any remaining favoured carbohydrates. Nevertheless, at 24 hpi *MIG1* and *TYE7* as well as *GAL4, SUC1, HMO1*, and *STD1* were significantly down-regulated. Despite increased expression of *MIG1* and *TYE7* early during infection, genes encoding major glycolysis enzymes such as *PFK1* were down-regulated during the entire infection (Fig. 5b). Instead, the transcription factor gene *STP2*, a regulator of amino-acid metabolism, showed increased expression early during infection. Consequently, expression of the amino-acid permease gene *GAP2* was increased throughout infection. In line with this, specific TCA cycle genes (*CIT1, ACO1, and MDH1*), genes in the glyoxylate shunt (*ICL1, MLS1*), and genes in the gluconeogenesis pathway (*PCK1, FBP1*) were up-regulated at 6 hpi (Fig. 5b).

To contextualize *C. albicans* metabolic reprogramming across multiple analysis levels, we overlaid our metabolome-driven in silico modelling predictions (Fig. S3B, Supplementary Data 2) with our transcriptome and metabolome data (Fig. 6, Table. S3). In silico analysis based on the metabolite availability at 12 hpi predicted reduced *C. albicans* glycolytic activity upon *L. rhamnosus* colonization, consistent with the down-regulation of transcription factors regulating carbon metabolism at 24 hpi, and key glycolysis genes at 6 and 24 hpi (Fig. 5b). In silico prediction also reflected most parts of the available transcriptional and metabolic data concerning the TCA cycle (Fig. 6). Both our metabolome and transcriptome data suggested relevance for the glyoxylate shunt, which *C. albicans* likely uses early during infection to bypass a full TCA cycle to use available amino acids as carbon sources to compensate for the absence of favoured nutrients. At 24 hpi *C. albicans* transcriptomic data suggested that the bias towards the glyoxylate shunt was lost and *C. albicans* tried to compensate by re-establishing a full TCA cycle and thus energy metabolism. In silico metabolic flux predictions at 12 hpi supported this notion and suggested decreased flux rates in citrate synthase and in succinate dehydrogenase opposed to increased flux in succinyl-CoA synthetase. Finally, several parts of the oxidative phosphorylation pathway were predicted to be less active in the presence of *L. rhamnosus*. XTT assays, which assess mitochondrial dehydrogenase activity, suggested a reduced mitochondrial metabolic activity when *C. albicans* is cultured in supernatants of *L. rhamnosus*-colonized IECs (Fig. S7A) or in *L. rhamnosus*-conditioned medium independent of host cells (Fig. S7B). In addition, two of the metabolites observed after *L. rhamnosus* colonization reduced mitochondrial dehydrogenase activity (Fig. S7C).

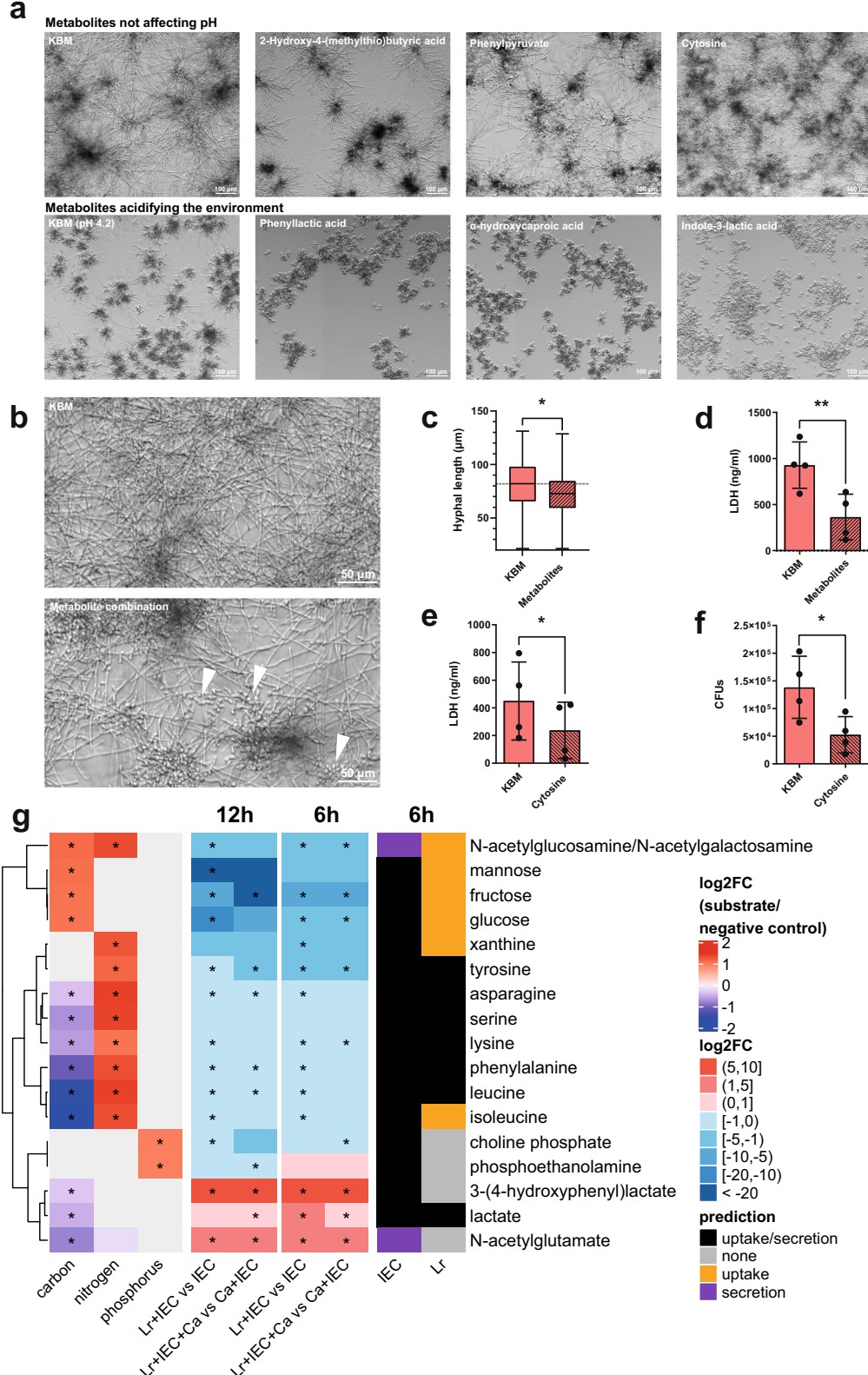

This was supported by the transcriptional data suggesting a trend towards an overall down-regulation of genes in complex 1 and 4 of the oxidative phosphorylation pathway (Figs. S7D, 6).

Our metabolome and transcriptome data combined with in silico metabolic flux predictions (Figs. 3g, S5) suggest that *C. albicans* undergoes drastic metabolic adaptations in response to

the suboptimal nutritional environment induced by *L. rhamnosus* colonization.

**Dysregulation of *C. albicans* virulence-relevant genes**. Changes in fungal energy metabolism could be linked to reduced pathogenicity. Decreased *C. albicans* damage potential in the presence

**Fig. 3 *L. rhamnosus* induces an unfavourable environment for *C. albicans*. a, b** Representative images of *C. albicans* morphology following growth in presence of (**a**) different metabolites at 50 mM, at neutral or acidic pH, for 20 h at 37 °C with 5% $CO_2$ ($n = 2$ biological replicates) or (**b**) a combination of selected metabolites each at 5 mM (cytosine, phenylpyruvate, 2-hydroxy-4-(methylthio)-butyric acid, 3-phenyllactic acid, 2-Hydroxyisocaproic acid and D-indole-3-lactic acid) and lactic acid at 15 mM, for 20 h at 37 °C with 5% $CO_2$ ($n = 4$ biological replicates). Yeast morphology is indicated with arrowheads. **c** Hyphal length of *C. albicans* grown for 4 h in KBM in the presence or absence of the combination of selected metabolites at 37 °C with 5% $CO_2$ ($n = 400$ cells examined over 4 independent experiments) (* = $p$ 0.0108). **d** Necrotic damage of IECs measured by the LDH activity in the supernatant at 24 hpi with *C. albicans* in the presence or absence of the combination of selected metabolites (** = $p$ 0.0093). **e** *C. albicans*-induced necrotic damage of IECs measured by the LDH activity in the supernatant (* = $p$ 0.0385) and (**f**) *C. albicans* translocation across the epithelial barrier assessed in the presence or absence of cytosine at 50 mM at 24 hpi (* = $p$ 0.0386). Bars represent the mean and SD of $n = 4$ independent experiments, dots represent the mean of the technical replicates of the individual experiments, boxplots represent the distribution of the total measurements (centre line, median; box limits, upper and lower quartiles; whiskers, range). Biological repeats were compared for significance using paired $t$ tests (two-tailed, one-sample) on the means of the technical replicates. Source data are provided as a Source Data file. **g** Phenotypic microarray growth experiments for wild-type *C. albicans* in presence of each metabolite as a carbon, nitrogen, or phosphorous source (left), metabolome data measured at 6 and 12 h and metabolic modelling predictions (right) are indicated for selected metabolites. For metabolic modelling, media was adapted from metabolome data derived from supernatants of IECs. Uptake or secretion was determined by identifying feasible flux ranges for metabolite-specific exchange reactions alongside optimization for biomass. Asterisks show statistical significance. ANOVA was performed for phenotypic microarrays (two-sided), Wilcoxon test for metabolomics (two-sided), with FDR correction. * = $p \le 0.05$. For the entire panel of metabolites see Fig. S5.

of *L. rhamnosus* could be mediated by the differential regulation of genes required for metabolic adaptation when these are linked to virulence. Therefore, 70 *C. albicans* mutants from a gene deletion mutant library[40] (Table S4), corresponding to a range of differentially regulated genes, were tested for their damage potential. The mutants *kre5Δ/Δ*, *ptp3Δ/Δ*, *orf19.4292Δ/Δ*, *ahr1Δ/Δ*, and *ace2Δ/Δ* were attenuated in IEC damage potential (Fig. 7a). The *ptp3Δ/Δ* mutant additionally showed reduced growth (Fig. 7b) and impaired filamentation (Fig. 7c).

In addition, *ypt7Δ/Δ*, *orf19.4459Δ/Δ*, *hyr1Δ/Δ*, *opt7Δ/Δ*, *rbe1Δ/Δ*, *pdk2Δ/Δ*, *orf19.7328Δ/Δ*, *zcf27Δ/Δ*, and *rgs2Δ/Δ* showed a hyper-damaging phenotype (Fig. 7a). Comparison of expression and mutant damage potential revealed a correlation between down-regulation and a reduced damage capacity of mutants of the corresponding genes with the exception of *kre5Δ/Δ*. Except *ypt7Δ/Δ* and *orf19.7328Δ/Δ*, genes corresponding to the hyper-damaging mutants were significantly up-regulated by *L. rhamnosus* at 6 or 12 hpi, suggesting that these represent potential antivirulence genes[41]. Collectively, this supports the hypothesis that the transcriptionally-regulated metabolic adaptions to *L. rhamnosus*-colonized IECs compromises the expression of virulence and antivirulence genes, thereby reducing the pathogenic potential of *C. albicans*.

## Discussion

Here, we investigated metabolic and molecular aspects of *L. rhamnosus*-mediated protection against *C. albicans* pathogenicity. While colonization with *L. rhamnosus* reduced the number of *C. albicans* in contact with the epithelium[26], reduced inocula still cause significant damage, hinting at additional mechanisms mediating the protection. We discovered that this contact-independent protection was associated with *Lactobacillus*-induced changes in the metabolic environment. These metabolic changes forced *C. albicans* to transcriptionally reprogram its metabolism, and these transcriptional changes were intertwined with genes required for pathogenicity.

*Lactobacillus* species shape the intestinal environment by consuming and releasing metabolites[42–45]. Antibiotic treatment reduced short chain fatty acid and secondary bile salt levels, while increasing carbohydrates and sugar alcohols, which enhanced *C. albicans* filamentation and colonization rates in mice[46–49]. We demonstrate that *L. rhamnosus* colonization not only creates an antagonistic environment, but also maintains it over time. Consistent with the rapid changes in the metabolic environment upon disturbances of the microbiota in vivo[46], a loss of the protective

effect was observed upon killing the bacteria with antibiotics even after *C. albicans* infection.

Metabolically active, proliferating *L. rhamnosus* cells are required for the antagonistic effect towards *C. albicans*[26], but the cell culture medium alone does not support *L. rhamnosus* growth. Combined in silico genome-scale metabolic simulations and in vitro experiments have been used before to shed light on the role of gut microbiota towards host metabolic disease[39,50]. Here, analysing both in conjunction revealed that epithelium-derived metabolites foster *L. rhamnosus* growth by providing nutritional support in specific pathways. This underscores that the host metabolic activity is required to support *L. rhamnosus* growth and its antagonistic effects in our model. The interplay of consumption of IEC-derived metabolites and subsequent metabolic activity of *L. rhamnosus*, highlights that *L. rhamnosus* may be metabolically specialized to survive in the presence of epithelial cells. Such interactions have been described to play a key role in the human gut, where the intestinal epithelium provides metabolites that selectively sustain beneficial members of the microbiota[51].

*L. rhamnosus* colonization of epithelial cells drastically changed the metabolic environment, an observation that can also be made in *L. rhamnosus*-colonized gnotobiotic mice[45]. We observed that specific antifungal and antivirulence compounds were secreted. *Lactobacillus* species are well known for their production of antimicrobial metabolites, which have been extensively studied outside the context of the host. Several metabolites that we detected upon *L. rhamnosus* colonization were previously characterized for their antifungal and antivirulence potential including: phenyllactic acid, mevalonolactone, 2-hydroxyisocaproic acid and 3-hydroxyoctanoate[35–38]. However, the recently characterized filamentation inhibiting metabolite β-carboline[21], was not detected in our metabolome, yet this may be attributable to the culture conditions or its degradation by host cells. We further identified several metabolites that affect *C. albicans* filamentous growth alone and in combination. The metabolite cytosine, which was observed to induce a hyphae-to-yeast transition, was validated for its antivirulence properties, in terms of reducing epithelial damage and translocation. Although the underlying mechanisms of the effects caused by the individual metabolites are not yet understood, it underscores that metabolic interplay between *C. albicans* and antagonistic bacteria contributes to promoting *C. albicans* commensalism.

In addition, *L. rhamnosus* colonization changed nutrient availability. Comparative analysis of metabolite availability from this study with previously published metabolic phenotyping of *C.*

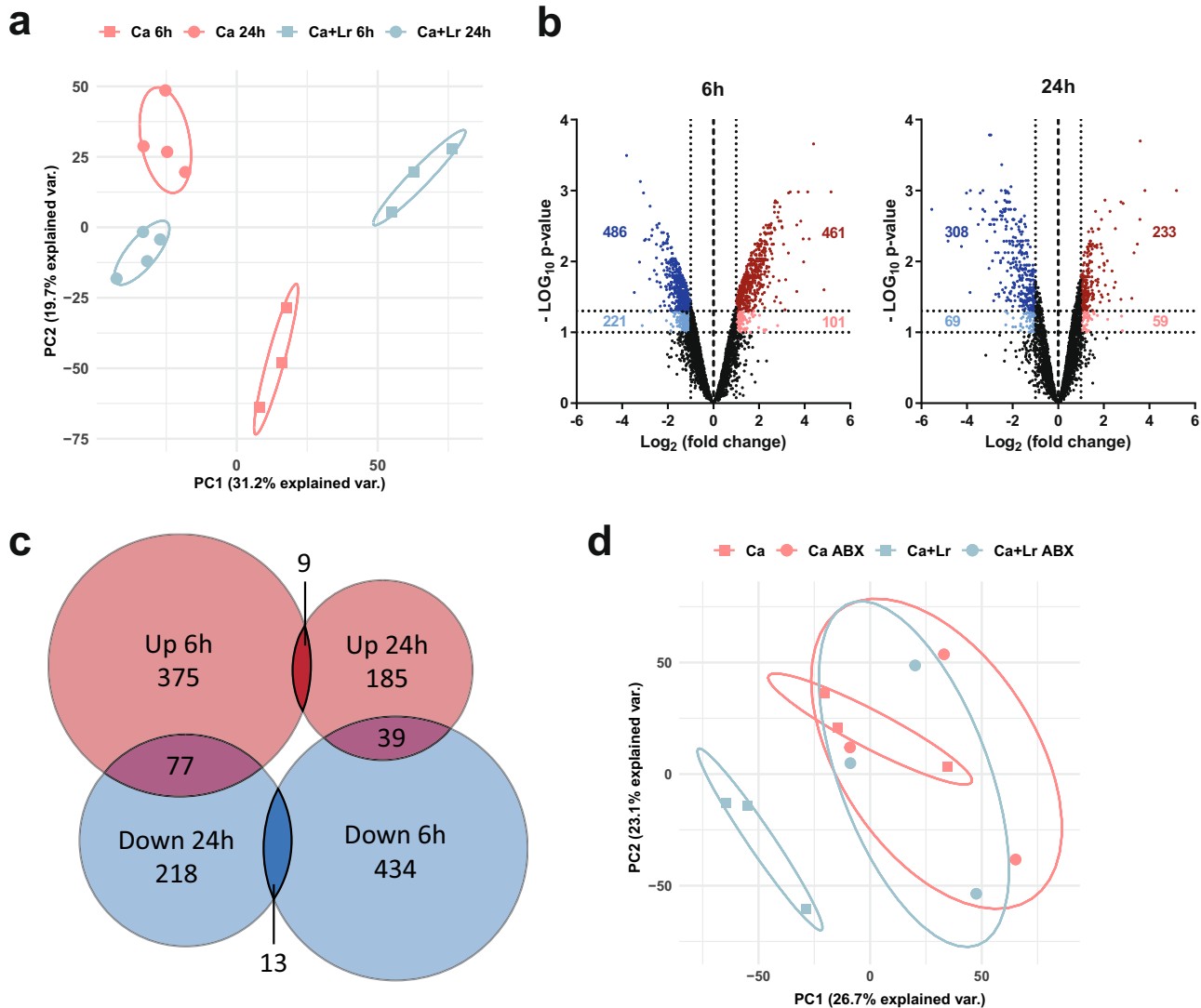

**Fig. 4 _C. albicans_ undergoes transcriptional changes during infection of _L. rhamnosus_-colonized IECs. a** Principal component analysis of _C. albicans_ gene expression at 6 and 24 h during in vitro infection of IECs in the presence and absence of _L. rhamnosus_ colonization. **b** Volcano plots showing differentially regulated _C. albicans_ genes at 6 and 24 hpi as a result of _L. rhamnosus_ colonization prior to infection based on the criteria of a Log₂ fold change of >1 or < −1 and a Bonferroni-corrected two-tailed moderated t-test p-value of <0.05 (dark blue and dark red) and <0.1 (light blue and light red). Source data are provided as a Source Data file. **c** Venn diagram analysis of the overlap in differentially expressed genes at 6 and 24 hpi. Data summarized from n = 3 and n = 4 independent experiments at 6 and 24 hpi, respectively. **d** PCA of _C. albicans_ gene expression at 24 hpi during in vitro infection of IECs in the presence and absence of _L. rhamnosus_ colonization and in the presence and absence of antibiotics.

_albicans_[39] revealed a depletion of preferred carbon and nitrogen sources, such as amino acids or glucose, and an enrichment of less-favoured carbon sources, such as lactate or malate. The utilization of different carbon sources can drastically influence _C. albicans_ fitness and pathogenicity[52].

Even though single metabolites can potently inhibit _C. albicans_ pathogenicity mechanisms such as filamentation, we believe that the promotion of commensalism by the bacterial microbiota is multifactorial. Both the production of antivirulence metabolites as well as alterations in the metabolic environment may equally promote commensalism and even depend on each other. Moreover, _L. rhamnosus_ has been described to produce chitin degrading proteins[16] and exopolysaccharides[17] that can inhibit hyphal morphogenesis. Likely a combination of metabolic antagonism and specific effector functions underlie the potent pathogenicity inhibiting effects of _L. rhamnosus_.

While our study only includes a single bacterial member of the microbiota, we believe that similar metabolic interactions and

competition by additional antagonistic bacteria underlie the strong association between a healthy microbiota and commensalism. Moreover, multiple antagonistic bacterial species may act synergistically in promoting _C. albicans_ commensalism.

Although we followed a one-model-at-a-time simulation approach, our in silico analysis revealed changes in key metabolic pathways and regulator genes of _C. albicans_, which we also found in our _C. albicans_ transcriptome data. Further, more sophisticated community modelling simulating all three GEMs simultaneously were largely in agreement with our single GEM simulations, but may be investigated in more depth in future work.

Several studies suggest that _C. albicans_ metabolism and virulence are interconnected[53–56], similarly to other fungal pathogens[57,58]. A variety of proteins regulate metabolism as well as virulence, including Yck2[56,59], Tye7[60,61], Gal4[60], Mig1[62], Mig2[62], and Ace2[63,64]. Supporting our hypothesis that alternative carbon sources reduce _C. albicans_ pathogenicity, several down-regulated genes, especially in the late phase of infection, were

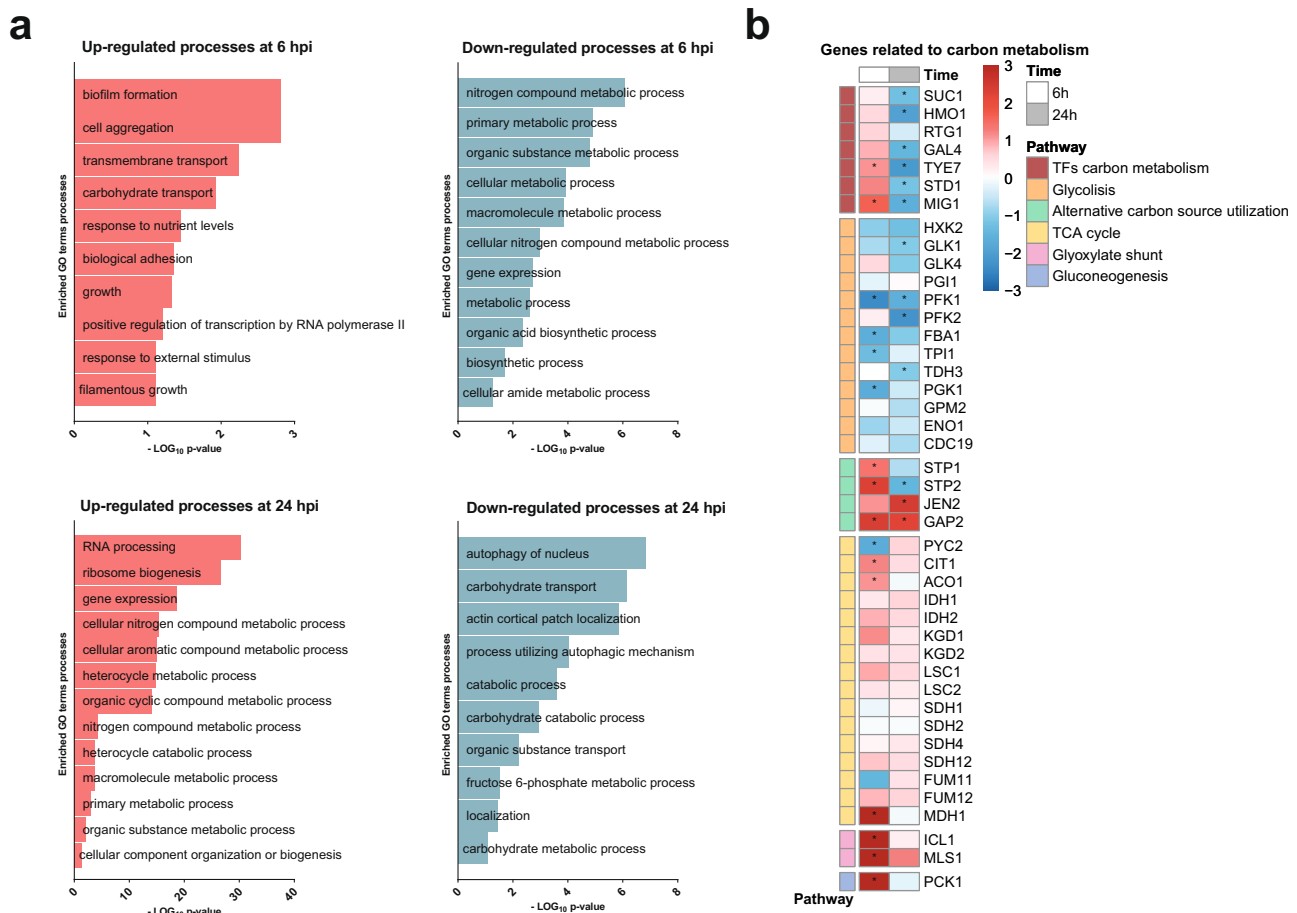

**Fig. 5 C. albicans undergoes transcriptional metabolic adaptations when infecting L. rhamnosus colonized epithelium. a** GO-term enrichment analysis of differentially regulated genes (Log2 fold change >1 or < −1 and p < 0.1) analysed with the GO-term finder on the Candida Genome Database website and reduced with the Revigo program (http://revigo.irb.hr/) (similarity: Tiny (0.4)). Significantly enriched GO terms are plotted based on the −Log10 p-value from the Bonferroni-corrected hypergeometric distribution. Data summarized from n = 3 at 24 hpi. Source data are provided as a Source Data file. **b** Heatmap highlighting the transcriptional regulation of C. albicans metabolic genes as a result of L. rhamnosus colonization at 6 and 24 hpi. Legend colour represents the Log2 fold change of the regulation in presence vs. absence of L. rhamnosus. The asterisks (*) represent significance, based on the criteria of a Log2 fold change of >1 or < −1 and a moderated t-test, Bonferroni-corrected two-tailed p-value of <0.05.

associated with carbohydrate metabolism[62]. Mig1, an essential regulator in the glucose repression pathway[62], was up-regulated early during infection of L. rhamnosus-colonized IECs, but down-regulated later, when only alternative carbon sources were available. However, existing studies demonstrated that a mig1Δ/Δ deletion mutant only showed attenuated virulence when MIG2 was simultaneously deleted[62]. Similarly, the deletion of the glycolysis-regulating genes TYE7 and GAL4, only showed attenuated virulence in a Galleria melonella infection model when a corresponding double deletion mutant (gal4Δ/Δ/tye7Δ/Δ) was investigated[60]. Both genes were down-regulated at later time points in our model.

Based on this, we believe that individual deletion or down-regulation of these metabolic transcription factors can be compensated by redundancy, which secures a high level of metabolic flexibility for C. albicans. However, when more than one of these genes are not expressed C. albicans loses its metabolic flexibility, which is associated with reduced pathogenicity. In line with this, a closer look at the carbohydrate catabolism revealed the down-regulation of glycolysis-relevant genes and an up-regulation of several TCA cycle and glyoxylate shunt genes starting at 6 h. Interestingly, this matches the metabolic phenotype of ace2Δ/Δ[64]. However, the ace2Δ/Δ mutant has a large cell morphology defect[64], which is associated with attenuated virulence[63].

Nevertheless, the phenotype of this mutant suggests that transcription factors like Ace2 affect both virulence as well as metabolism.

Several genes of the oxidative phosphorylation pathway were suppressed upon L. rhamnosus colonization, such as ADH1 or COX2. Impaired oxidative phosphorylation could limit ATP production, which may reduce fungal growth, filamentation, and virulence. A link between respiration and fungal pathogenesis has been described[65] and specific correlations between C. albicans oxidative phosphorylation and pathogenicity mediated by ADH1 have been observed[66].

The impact of transcriptional changes on C. albicans virulence was validated using deletion mutants. Deletion mutants of the genes PTP3, AHR1, ACE2, and orf19.4292, which are down-regulated during infection of L. rhamnosus-colonized IECs, exhibited an attenuated damage potential. Moreover, ptp3Δ/Δ was compromised in growth and filamentation. PTP3 encodes a protein tyrosine phosphatase required for hyphal maintenance[67]. The transcription factor Ahr1 regulates the virulence genes ALS3 and ECE1[68], but can also repress the white-to-opaque switch[69]. Orf19.4292 encodes the prevacuolar trafficking protein Pep12, which is essential for virulence in mice[70]. The transcriptional regulator Ace2 regulates glycostress metabolism[64] and its role in virulence is discussed above. Our data underscores that down-

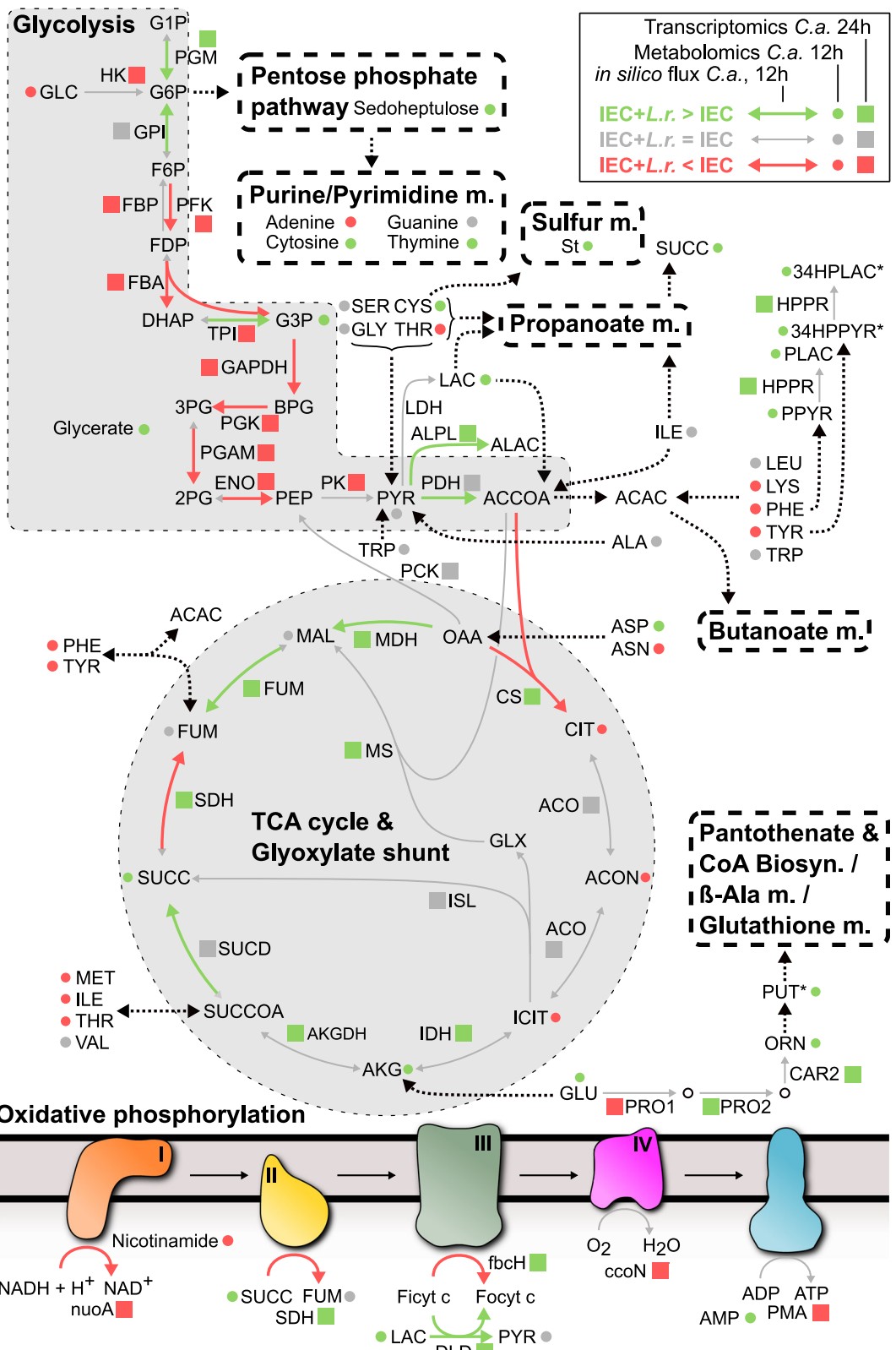

**Fig. 6 Central metabolism of *C. albicans* is altered by *L. rhamnosus* colonization.** Reactions associated to glycolysis, TCA cycle, oxidative phosphorylation are indicated, as well as relationships with additional metabolic pathways (pentose phosphate pathway, nitrogenated bases, sulfur metabolism, butanoate, propanoate, and pantothenate, CoA, β-Alanine and glutathione metabolism). Information on metabolome (12 h) and transcriptomic (24 h) data are combined with in silico genome-wide metabolic flux predictions (12 h). Dotted arrows represent several combined reactions. For the reaction abbreviations see Table S3.

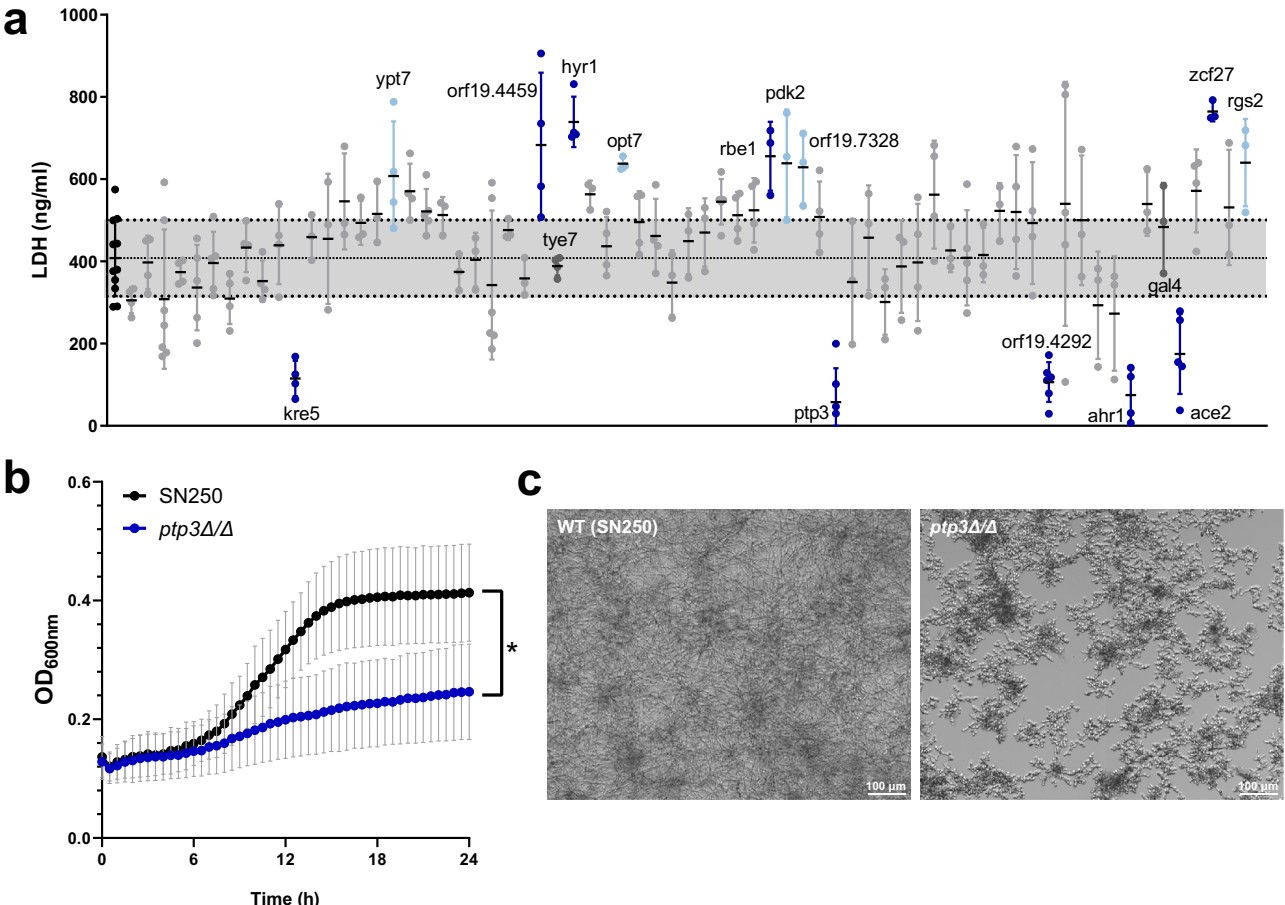

**Fig. 7 Screening of *C. albicans* deletion mutants. a** Ability of deletion mutants to induce necrotic cell damage of IECs assessed by LDH activity in the supernatant at 24 hpi. Data are shown as the mean and standard deviation (SD) with dots showing the individual replicates (*n* = 3–11 biological replicates). Deletion mutants were compared to the wild-type control using a one-way ANOVA and Dunnett's Multiple Comparison post-hoc analysis. Mutants with a significantly increased or decreased damage potential (dark blue, *p*-value ≤ 0.05; light blue, *p*-value ≤ 0.1) are labelled (see Supplementary Table 4 for exact *p*-values). Horizontal lines correspond to the mean of the damage induced by the wild-type ± SD. **b** Growth rates of the *ptp3Δ/Δ* mutant with significantly reduced growth (blue line) compared to the parental strain (black line) in KBM. Lines represent the mean and SD of *n* = 3 independent experiments and were compared for significance using a two-way Repeated Measures ANOVA, * = *p* 0.0188. Source data are provided as a Source Data file. **c** Representative images of the *ptp3Δ/Δ* deletion mutant and the parental strain morphologies, after 24 h incubation in KBM at 37 °C with 5% $CO_2$.

regulation of each of these genes could be sufficient to reduce pathogenicity. This supports the notion that *C. albicans* metabolic and transcriptional adaptations upon infection of *L. rhamnosus*-colonized IECs are intertwined with pathogenic potential.

Other studies also observed changes in the *C. albicans* gene expression induced by lactobacilli. During oral epithelium infection, *C. albicans*-induced expression of genes associated with diverse metabolic pathways[71]. *L. plantarum*, *L. helveticus*, and *L. crispatus* also down-regulated hypha-associated genes (HAGs)[72,73]. As the majority of HAGs were not affected in our dataset, we hypothesize that niche-specific modes of action exist for the diverse *Lactobacillus* species.

Collectively, our results demonstrate that protection by *L. rhamnosus* colonization is a multifactorial process that synergistically affects *C. albicans* growth and pathogenicity. Different aspects of this complex interaction have been individually assessed in the past. Here we provide a multilevel comprehensive picture of the interplay between *C. albicans* and its antagonist *L. rhamnosus* in a human, gut-like experimental setup. The metabolic and transcriptional insights into the antagonistic potential of a single member of the microbiota underline the importance and complexity of a balanced intestinal microbiota that keeps *C. albicans* in its commensal state.

Our combined metabolome data, in silico metabolic modelling, transcriptome and mutant screening, as well as in vitro validations, provide fundamental, contextualized insights into how *C. albicans* pathogenicity can be controlled or prevented.

## Methods

**In vitro model and infection**. An in vitro intestinal *C. albicans*-infection model was used to perform the experiments as previously described[26]. C2BBe1 (ATCC CRL-2102) and HT29-MTX (ATCC HTB-38; CLS, Lot No. 13B021) cells were seeded in collagen I coated (10 µg/ml, 2 h at room temperature [RT]; Thermo Scientific) 6-well, 24-well or 96-well plates at a ratio of 70:30 with a total cell density of $4 \times 10^5$ cells/well (6-well), $1 \times 10^5$ cells/well (24-well) and $2 \times 10^4$ cells/well (96-well, transwell). Cells were used for experiments after 14 days of differentiation in Dulbecco's Modified Eagle's Medium (DMEM; Gibco, Thermo Scientific) supplemented with 10% foetal calf serum (FCS; Bio & Sell), 10 µg/ml Holotransferrin (Calbiochem, Merck), and 1% non-essential amino acids (Gibco, Thermo Scientific) with medium exchange three times per week. Cell lines have been authenticated via commercial STR profiling on November 29, 2019 (Eurofins Genomic) and checked for mycoplasma contaminations using a PCR mycoplasma test kit (PromoKine) according to the manufacturer's instructions.

For colonization, DMEM was exchanged for 1.3 ml (6-well) or 50 µl (96-well) serum-free Keratinocyte Basal Medium (KBM) (Lonza, Basel, Switzerland) and monolayers were colonized with 1.3 ml (6-well), 250 µl (24-well) or 50 µl (96-well) *Lactobacillus rhamnosus* ATCC 7469 (OD 0.2 in KBM) for 18 h prior to infection. For contact-independent colonization, DMEM was exchanged for 600 µl in the bottom of a 24-well plate and a transwell insert with 125 µl *L. rhamnosus* (OD 0.4 in KBM) was placed on top.

Fungal infection was performed with 1.3 ml (6-well), 250 μl (24-well) or 50 μl (96-well) *C. albicans* WT SC5314 ($4 \times 10^5$ cells/ml in KBM). For the samples with antibiotic treatment, 500 μg/ml Gentamicin (Merck) and 1× PenStrep (Gibco, Thermo Scientific) were added 4 hpi. For infection in presence of metabolites, a solution of each metabolite in KBM was prepared fresh and sterile filtered. Then, concentration was adjusted and 50 μl (96-well, transwells) of this solution with *C. albicans* (MOI 1) were used to infect the cells for 24 h. Wells with only medium, *L. rhamnosus*, or *C. albicans* in the presence or absence of the host cells served as controls. Infected cells and controls were incubated at 37 °C with 5% $CO_2$. Data from in vitro damage, fungal burden, and translocation assays were analysed using GraphPad prism version 8. Data from at least 3 biological replicates were analysed for statistical significance using a t-test or a one-way ANOVA with multiple comparisons. Statistical significance is depicted in the figures: * = $p \leq 0.05$, ** = $p \leq 0.01$, or *** = $p \leq 0.001$.

***C. albicans* CFU quantification**. To determine how many *C. albicans* cells were present 24 hpi, CFU quantification was performed in 96-well plates. Supernatants were collected and IECs were treated for 5 min with 0.2% Triton-X-100 (Sigma-Aldrich) to lyse the host cells and release adherent fungal cells. After detaching adherent host cells via scraping with a pipette tip, the lysate was added to the respective supernatant. Wells were washed twice with PBS. The final samples were diluted appropriately with PBS and plated on YPD agar with 1× PenStrep (Gibco, Thermo Fisher Scientific) and incubated at 30 °C until adequate growth for CFU counting was reached (24 h).

**Quantification of cytotoxicity (LDH)**. The host-cell damage was determined by measuring the activity of cytoplasmic LDH[74] (Fig. S8). LDH activity was quantified in the supernatant of infected IEC monolayers in 96-well plates 24 hpi using the Cytotoxicity Detection Kit (Roche) according to the manufacturer's instructions. LDH from rabbit muscle (5 mg/ml, Roche) was used to generate a standard curve for the determination of LDH concentrations. The background LDH activity control level of uninfected IECs was subtracted from the test conditions.

**Metabolome analysis**. Supernatants for untargeted metabolomics were collected 6 and 12 h after *C. albicans* infection (Fig. S8). 500 μl of the supernatant was collected from 6-well plates, centrifuged, snap-frozen in liquid nitrogen, and stored at −80 °C until analysis. Samples were analysed and interpreted by Metabolon (Morrisville, US). Experiments included three technical replicates and five independent experiments were performed in total.

Raw metabolome data was rescaled to set the median equal to 1, and the missing values imputed with the minimum. Data were loaded in R version 1.2.5019[75], rows were normalized and Euclidian distances were calculated. The heatmap was generated with the "pHeatmap" package v1.0.12[76] and used to manually obtain the optimal number of clusters. The distance matrix was subjected to hierarchical cluster analysis using the complete linkage agglomeration method and metabolites were classified according to their cluster. Colour bars indicating cluster or condition were added to the dendrograms using the R package "dendextend" v 1.15.2[77]. Metabolite enrichment analysis were conducted in MetaboAnalyst 5.0[78] across KEGG pathways using overrepresentation analysis (ORA; hypergeometric test). All results across all mentioned clusters are presented and significant hits are indicated. FDR correction was done per cluster and time point, and FDR ≤ 0.1 was considered. Proportional Euler diagrams were done using the R package "eulerr"[79].

**L. rhamnosus growth**. *L. rhamnosus* was cultured in MRS Broth for 48 h at 37 °C with 5% $CO_2$ and 1% $O_2$. Afterwards, cells were washed, and 3000 cells/ml were inoculated in the different media and IECs were colonized. At 0, 24, and 48 h independent wells were resuspended, and the content was appropriately diluted and plated on MRS agar. Plates were incubated for 48 h at 37 °C with 5% $CO_2$ and 1% $O_2$ until CFU quantification. Tested metabolites are shown in Table 1.

**Live-cell imaging**. *C. albicans* ($1 \times 10^4$ cell/ml) in KBM with or without supplementation of individual metabolites or in combination was incubated for 24 h at 37 °C with 5% $CO_2$ inside the Cell Discoverer 7 microscope (Zeiss), in which a bright field picture was taken every hour, images were exported with Zeiss Zen3.1 (blue edition). Hyphal length was measured at 4 h images using Zeiss Zen3.1 (blue edition) Table 2.

**Translocation assay**. To determine translocation through the epithelial barrier, infections were performed in transwell inserts (Sarstedt) with a pore size of 5 μm. 24 hpi, zymolyase (260 U/ml) was added to the bottom compartment and incubated for 2 h at 37 °C with 5% $CO_2$. Afterwards, samples were diluted in PBS, plated on YPD agar, and incubated at 30 °C for 24 h.

**Transcriptional profiling**. After 6 and 24 h of *C. albicans* infection in 6-well plates, RNA isolation of *C. albicans* was performed (Fig. S8). At the appropriate time points, 650 μl RLT buffer was added to the wells and the plates were frozen in liquid nitrogen immediately. After thawing, fungal and host cells were collected via

**Table 1 Metabolites tested to for their support of *L. rhamnosus* growth in KBM.**

| Compound name | Experimental concentration | Company |
|---|---|---|
| Citric acid | 5 mM | Roth |
| Carnitine | 5 mM | Sigma |
| Gamma-glutamyl-alanine | 5 mM | Sigma |
| Xanthine | 0.05 mM | Sigma |
| 3-methyl-2-oxovalerate | 5 mM | Sigma |
| 3-methyl-2-oxobutyrate | 5 mM | Sigma |
| Non-Essential Amino Acids + L-Glutamine | 1 mM | Gibco |
| Fructose | 1 mM | Sigma |
| Nicotinamide | 0.001 mM | Fluka |
| N-acetyl-glucosamine | 1 mM | Sigma |
| Carnosine | 5 mM | Sigma |

**Table 2 Tested metabolites to impair *C. albicans* filamentation.**

| Compound name | Experimental concentration | | Company |
|---|---|---|---|
| | alone | in combination | |
| 2-deoxyinosine | 50 mM | – | Sigma |
| Allantoine | 50 mM | – | Roth |
| Alpha hydroxycaproate (HICA) | 50 mM | 5 mM | Sigma |
| Cytosine | 50 mM | 5 mM | Sigma |
| Histidine | 50 mM | – | Roth |
| Hydroxymethylbutyrate | 50 mM | 5 mM | Sigma |
| Indolelactate | 50 mM | 5 mM | Sigma |
| Sodium lactate | 50 mM | 15 mM | Sigma |
| Phenyllactic acid | 50 mM | 5 mM | Sigma |
| Phenylpyruvate | 50 mM | 5 mM | Sigma |
| Pipecolate | 50 mM | – | Sigma |
| Thymine | 50 mM | – | Sigma |
| Uridine | 50 mM | – | Roth |

scraping. The collected material was centrifuged and fungal RNA isolation was performed on the pellet according to a previously described protocol[80]. RNA quantities were determined with a NanoDrop 1000 Spectrophotometer (Thermo Fisher Scientific) and RNA quality was verified with an Agilent 2100 Bioanalyzer (Agilent Technologies). RNA was subsequently converted into Cy5-labeled cRNA (Cy5 CTP; GE Healthcare, United Kingdom) using a QuickAmp labelling kit (Agilent). Samples were co-hybridized with a common Cy3-labeled reference (RNA from mid-log-phase-grown *C. albicans* SC5314[81]) on Agilent arrays (*C.a.*: AMA-DID 026869), scanned in a GenePix 4200AL with GenePix Pro 6.1 (Auto PMT; pixel size, 5 μm). Differentially expressed genes (DEGs) (p-value: 0.05; $Log_2$ fold change) were analysed with GeneSpring 14.9 (Agilent) and the *Candida* Genome Database (CGD; http://www.candidagenome.org). Out of the 6130 *C. albicans* genes, 5125 genes were used for the analysis after filtering on the minimal fluorescent signal and subtraction of the background signal. Genes were considered differentially regulated when they had a moderated T-test, Bonferroni-corrected p-value of <0.05 and a $Log_2$ fold change of more than 1 or less than −1.

Gene expression data were exported from GeneSpring and loaded in R version 3.6.2[75], rows were normalized and Euclidian distances were calculated. The heatmap was generated with the "pHeatmap" package v1.0.12[76] and the distance matrix was subjected to hierarchical cluster analysis using the complete linkage agglomeration method. Colour bars indicating condition were added to the dendrograms using the R package "dendextend" v 1.15.2[77]. The PCA was calculated using the R function "prcomp". Graphs were generated using the "ggbiplot" package v 0.55[82]. Proportional Euler diagrams of the DEGs were done using the R package "eulerr"[79] and "eulerAPE"[83].

GO term enrichment of differentially expressed genes was analysed using the GO-Term Finder on the *Candida* genome database[84], which uses a hypergeometric distribution with Multiple Hypothesis Correction (Bonferroni Correction) to calculate p-values. Subsequently, the significantly enriched GO-terms were processed using REVIGO[85] (similarity: Tiny (0.4); database: whole Uniprot; semantic similarity measure: SimRel) to remove overlapping and redundant GO-terms.

**Bioinformatics**. GEMs for *C. albicans*, human, and *L. rhamnosus* metabolism were used to simulate and analyse different growth scenarios in silico. Specifically, the recently published model for *C. albicans*[39] was downloaded from the supplementary material of the publication. The GEMs for *Lactobacillus rhamnosus* LMS2-1 for *L. rhamnosus*[32], and Recon3D 3.01[31], a comprehensive generic GEM of human metabolism used to simulate human intestinal epithelial cells, were downloaded from www.vmh.life. Metabolomics data were used to modulate feasible nutrition uptake for each model via respective exchange reactions as defined by each GEM. Feasible uptake rates for available metabolites were adapted from the metabolome measurements across all investigated conditions. Feasible uptake flux ranges for each metabolite in our GEMs were kept in the range [0, 1000] mmol/g(DW)h. The metabolite concentrations for each sample were transformed into this range based on the metabolite glutamine showing the highest concentration in the 12 hpi *L. rhamnosus* supernatant compared to all measured metabolites and all samples. The uptake rate of glutamine was set to 1000 mmol/g(DW)h accordingly, whereas all others were set to the respective proportion to the maximum glutamine value. The biomass function of each GEM was used as objective function for all metabolic modelling simulations. To obtain objective function values mimicking an anaerobic environment (oxygen influx prohibited) as well as feasible reaction flux ranges supporting at least 90% of the objective function flux, we applied flux balance analysis (FBA) and flux variability analysis (FVA) across all tested conditions for all tested GEMs[30,33]. All GEM analyses were done in COBRApy[86] using Python 3.6.4 and the IBM ILOG CPLEX Optimizer (version 12.8).

**Mitochondrial activity assessment with XTT**. To obtain the *L. rhamnosus*-conditioned supernatants, a culture of *L. rhamnosus* was washed and adjusted to an $OD_{600}$ of 0.2 in KBM, and 1 ml was used to colonize a confluent layer of differentiated intestinal epithelial cells (C2BBe1:HT29-MTX, 70:30) for 24 h (37 °C, 5% $CO_2$). *L. rhamnosus* was also adjusted to an $OD_{600}$ of 0.2 in KBM + (KBM supplemented with 5 mM citric acid, 5 mM carnitine and 5 mM gamma-glutamyl-alanine) and incubated in a 24-well plate independently of host cells for 24 h (37 °C, 5% $CO_2$). After 24 h, conditioned media were filtered. *C. albicans* was washed and adjusted to a concentration of $1 \times 10^5$ or $1 \times 10^6$ cell/ml in the different media and incubated in a 96-well plate for 1 h (37 °C, 5% $CO_2$). After that time, the plate was centrifuged and supernatants were exchanged with XTT reagent (PBS with 0.2 mg/ml XTT, VWR Life Sciences and 1.1 µg/ml menadione sodium bisulfite, Roth). The plate was then incubated for 2 h at 37 °C. Afterwards, 100 µl of the supernatants were transferred to a new plate and absorbance at 492 nm measured.

To compare XTT absorbance to biomass, a crystal violet assay was performed afterwards. The supernatant from each well was removed and the plate was left to air dry. Afterwards, 150 µl crystal violet 1% (Sigma) was added. After 45 min incubation at room temperature, crystal violet was removed, and the wells were washed three times with distilled water. Crystal violet was then solubilized with 200 µl ethanol 99% for 45 min. Afterwards, 100 µl of the supernatants were transferred to a new plate and absorbance was measured at 550 nm (Tecan Infinite M200; i-control software).

The same procedure was used to assess the effect of the metabolites in the mitochondrial activity, except the preincubation with the metabolite was done for 24 h prior to the XTT assay and the initial *C. albicans* concentration was $1 \times 10^4$ cell/ml.

**Deletion mutant screening**. *C. albicans* mutants from the deletion mutant library[40] (Tab. S4) were cultivated in YPD broth in 96-well plates and incubated overnight at 30 °C with shaking at 180 rpm. The overnight cultures were adjusted to an $OD_{600}$ of 0.0025 in KBM. Then, the diluted overnight cultures were diluted 1:2 in KBM in 96-well plates with a confluent layer of differentiated intestinal epithelial cells (C2BBe1:HT29-MTX, 70:30). Damage was measured via the LDH activity assay (see Quantification of cytotoxicity (LDH)). Positive hits were validated with additional damage experiments where the mutants were cultivated in YPD broth in 25 ml Erlenmeyer flasks overnight at 30 °C with shaking at 180 rpm. Fungal cells where then washed in PBS, counted, and adjusted to a concentration of $4 \times 10^5$ cells/ml in KBM. The epithelial cell monolayer was infected with 50 µl (96-well, transwell) or 250 µl (24-well). For growth curves, the diluted overnight cultures were diluted 1:2 in KBM in 96-well plates and incubated for 24 h at 37 °C with 5% $CO_2$ in a microplate reader (Tecan Infinite M200; i-control software). Growth was monitored with $OD_{600}$ measurements every 30 min over 24 h. Data was analysed using Graphpad prism version 8. Data from at least 3 biological replicates was analysed for statistical significance using a one-way ANOVA with multiple comparisons.

**Reporting summary**. Further information on research design is available in the Nature Research Reporting Summary linked to this article.

## Data availability
The authors declare that the data supporting the findings of this study are available within the paper and its Supplementary Information files. The source data are provided as Source Data file. The transcriptomics data generated in this study has been deposited in the Array Express database under accession code: E-MTAB-11090. The untargeted metabolomics data generated in this study are provided in the Supplementary Data 1. The biomass objective function values and associated flux ranges for all reactions for all simulations and investigated media conditions generated in this study are provided in the Supplementary Data 2. The simulation sets over different fractions of required objective function values generated in this study are provided in the Supplementary Data 3. Databases used in this study include: *Candida* Genome Database (http://www.candidagenome.org/) GO-term finder, RECON3d human metabolic model (www.vmh.life/files/reconstructions/Recon/3D.01/Recon3D_301.zip).

## Code availability
The python code for the metabolic models and simulations[87] can be found in the following repository: https://doi.org/10.5281/zenodo.6501838.

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

## Acknowledgements

This work was funded by the Deutsche Forschungsgemeinschaft (DFG, German Research Foundation) under Germany's Excellence Strategy—EXC 2051—Project-ID 390713860 (R.A.R., B.H.). M.S.G. was supported by the German Research Foundation (Deutsche Forschungsgemeinschaft—DFG) Emmy Noether Program (project no. 434385622/GR 5617/1-1). A.L., K.G. and B.H. were supported by the Infect ERA-NET Program FunComPath (BMBF 031L0001A) and the Centre for Sepsis Control and Care (CSCC, BMBF 01EO1002). J.L.S., B.H., M.H.M., P.G., G.P., S.S., S.V. and M.S.G. are further supported by the DFG within the Collaborative Research Centre (CRC)/Transregio (TRR) 124 "FungiNet" projects C1, C2, B5 and INF (DFG project number 210879364). S.M. and B.H. are further supported by a grant from the Wellcome Trust (215599_Z_19_Z). S.V. is further supported by the German Ministry for Education and Science in the program Unternehmen Region (BMBF 03Z22JN11). Authors would like to thank Dr. Sascha Brunke for his assistance in data management and providing Duplos.

## Author contributions

R.A.R. designed and performed experiments, interpreted data and results, drafted and reviewed the manuscript. A.L. performed experiments, interpreted the results, and drafted the manuscript. J.L.S., L.M., K.G. and R.G. performed experiments. P.G., M.H.M. and S.S. performed the in silico analysis; in addition, MHM and SS drafted and reviewed sections of the manuscript on the in silico analysis. S.V. contributed to data acquisition, provided intellectual support and supervision and reviewed the manuscript. G.P. and S.M. provided intellectual support and supervision and reviewed the manuscript. B.H. conceived the study, provided intellectual support and supervision and reviewed the manuscript. M.S.G. conceived the study, interpreted data, assisted in experimental design, drafted and reviewed the manuscript.

## Funding

## Competing interests

The authors declare no competing interests.
