## [Peer Review File · Nature Communications]

Reviewers' Comments:

Reviewer #1:

Remarks to the Author:

In this manuscript, Alonso-Roman and colleagues use "omics" to understand how the presence of a bacterium, *Lactobacillus rhamnosus*, reduces the ability of the fungus *Candida albicans* to cause epithelial cell damage.

It can be expected that bacteria and fungi antagonise each other by competing for nutrients and/or by secreting metabolites that impact on microbial growth, physiology, metabolism and/or development. Indeed, this study provides evidence for both of these possibilities. The novel aspects of the study include a comprehensive metabolomics and transcriptomics approach to profile a complex system consisting of all three relevant cell types (bacteria, fungi and host cells), showing that metabolites produced by host cells support bacterial growth enabling bacteria to protect them from *C. albicans*, the identification of cytosine's inhibitory effect on hyphal (but not yeast) growth of *C. albicans*, and the identification of several fungal mutants with reduced capacity to damage epithelial cells.

I appreciate that "omics" approaches will identify a complex picture, particularly in a biological scenario of co-culture of host, bacterial and fungal cells. As the authors conclude, this complexity might mean that a combination of several different mechanisms is involved in the inhibitory effects of *L. rhamnosus* on *C. albicans*. This study touches superficially on a few different mechanisms but does not explore any of the aspects in more depth. In my opinion, this is a weakness of the manuscript in its current form. Some additional in-depth analyses would be important to increase the value of the study beyond serving as a resource of the metabolic and transcriptional changes induced in *C. albicans* during co-culture with *L. rhamnosus* in the epithelial infection model.

Below, I am discussing some of the aspects that I found interesting and which could be explored more. Of course, not all of these questions can be addressed in one manuscript and I am not suggesting that they should be. But at least one of these aspects could be explored in more depth.

(i) It is not surprising that the bacteria would deplete the nutrients needed for *C. albicans*, causing a metabolic shift in the fungus. As the authors suggest, this explains the reduced growth of *C. albicans* in the presence of *L. rhamnosus*. A key result in my view is shown in Figure 1, showing that the reduction of *C. albicans*-induced damage by *L. rhamnosus* goes beyond inhibition of fungal growth.

If I am not mistaken, it has been reported that *Lactobacillus* limits *C. albicans* hyphal formation. Here the authors show that cytosine (secreted by *L. rhamnosus*) is inhibitory for invasive hyphal growth (importantly, yeast-morphology cells could grow in the presence of cytosine). This is very interesting, but the mechanism by which cytosine specifically inhibits hyphal growth is unclear. While a full characterisation of this mechanism is a whole new study and beyond the scope of here, some additional data would be beneficial. Are the concentrations of cytosine used in vitro (50mM) physiologically relevant? Have the authors tried any of the other metabolites secreted by *L. rhamnosus* for hyphal inhibition? In other words, is cytosine the key metabolite for hyphal inhibition?

(ii) The authors further suggest that the metabolic adaptation of *C. albicans* leads to differential expression of several genes that are required for causing epithelial cell damage (mutants analysed in Figure 6). This is interesting and re-enforces the idea that metabolism sits at the core of the adaptive programs that pathogens need to cause disease. However, the link between the metabolic reprogramming induced by *L. rhamnosus* and the differential expression of these genes remains unexplored. Transcriptomics identified several transcriptional and signalling regulators which could be explored further.

The authors selected the transcription factor *Ace2* for some further analyses, showing that the *ace2* mutant causes reduced epithelial cell damage compared to the wild type strain. It is suggested here that *Ace2* might be important for the observed metabolic regulation by *L. rhamnosus* because a study by Geraldine Butler's group (Mulhern et al 2006) showed that the *ace2*

mutant shows some of the same metabolic changes as seen in the presence of *L. rhamosus*. However, I am not sure if the effects of Ace2 on metabolism are direct.

Have the authors considered that the *ace2* mutant has a large cell morphology defect, which could be causative of the reduced host cell damage? In fact, Ace2 is best known as the activator of cell separation genes (the enzymes needed to degrade the cell wall following cell division). Therefore, the *ace2* mutant grows as cell chains and is likely to display cell wall changes that could impact invasion. The differences in cell morphology between the wild type and *ace2* could also be driving some of the metabolic changes. The Butler lab paper also suggested that the metabolic genes differentially expressed in the *ace2* mutant are not likely to be direct targets of Ace2, due to lack of Ace2 binding sites.

The Discussion mentions differential expression of several other transcription factors with links to carbon metabolism (Mig1, Tye7 and Gal4) –these factors likely have more direct roles in metabolic regulation than Ace2. Have they been tested for epithelial cell damage or capacity to reprogram fungal metabolism in the presence of *L. rhamosus*?

(iii) Transcriptome analyses suggested reduced mitochondrial respiration in the presence of *L. rhamosus*. Lower respiratory capacity could reduce hyphal growth of *C. albicans* (this has been shown by multiple labs), but some experimental corroboration of the respiratory inhibition by *L. rhamosus* would be required.

I was also thinking that if glycolysis is reduced in the presence of *L. rhamosus*, then respiration might be expected to be enhanced (not reduced). Do any of the inhibitory metabolites produced by *L. rhamosus* inhibit mitochondria?

(iv) The Results section on the transcriptomics data is in need of improvement. Neither Figure 4 nor Figure S5 go beyond high-level representation of the differentially expressed genes and GO enrichments. Further discussion and depiction of the specific metabolic genes that were affected, as well as the regulators would be valuable for the reader. For example, transcriptional regulators of carbon metabolism are discussed in the Discussion (lines 319-326) but not mentioned in the Results.

Reviewer #2:

Remarks to the Author:

The manuscript by Alonso-Roman et al investigates the interactions between *C. albicans*, *L. rhamosus*, and intestinal epithelial cells. Using the transcriptomics, metabolomics, metabolic modelling and reverse genetics, revealed antivirulence and antifungal metabolic activity. Moreover, they observed epithelial cells promote the *L. rhamosus* growth through specific metabolism and this showed affect the global metabolite profile available to the community and favoured *C. albicans* to metabolically reprogrammed which affect the virulence genes. I found the work timely and important considering the pathogenicity of *C. albicans*. The paper has generated in-vitro and in-silico data and integrate these data to support the findings. While I have found the results and the work timely and well presented, based on my acquired skills, I have few comments on the metabolic modelling section of the paper which I hope the authors find them constructive to improve their work, finding and making the analysis clearer.

The authors mentioned they have done untargeted metabolomics however selected 235 metabolites. Could please elaborate more how these 235 metabolites were chosen as usually untargeted metabolomics produced more domain of metabolites. Also, how the metabolite profile was calculated considering the initial media? In what stage of the growth the sampling for the metabolites were done?

-For the GEMs and metabolic modelling, the authors have referred to the already generated models and cites two papers. However, looking at VMH and also the RECON3D paper, there are no specific *C. albicans* and Epithelial models. I guess the human model has been used for the intestinal epithelial model. This needs to be clearly addressed that the human generic model has

been used otherwise there are other available tissue specific model that can be applied here such as <https://metabolicatlas.org/gems/repository>

-The simulation performed by each of the organisms, the constraints and objective functions have not specified in the paper and just mentioned in the methods "In brief, feasible uptake rates for available metabolites were adapted from the metabolome measurements across all investigated conditions." Please support this analysis with metabolite uptake and secretion and this analysis need to be supported as well with the microbial biomass productions and sensitivity analysis.

-How the conversion of the untargeted metabolites without unit of concentration to a flux unit for input has been done? Can you support this calculation and what is the assumption for the dilution rate? In the method it also mentioned the modelling were done anaerobically. Was this the case for the epithelial and Candida model?

-Has the author tried to perform pairwise or community modelling to simulate the interactions between the three organisms at once? This type of modelling can be more suitable for this study. There are already available functions in the COBRA to perform these analyses.

-Based on the method, the pathway enrichment analysis was done using Revigo. What type of statistics were performed to report the significant ones?

Reviewer #3:

Remarks to the Author:

The authors are to be congratulated on what is an elegant study that advances our knowledge of host/microbe interactions as well as providing what I see as a template for future studies of this nature. The paper is very well written and the data are analysed in a manner that fully supports the statements made within the manuscript. The data are of a high quality and the rationale for experimental design is very clear.

Although I believe the paper could be accepted as it is now I would make some small suggestions as points for consideration.

I feel it would be better to include FigS1A in place of Figure 1A as it contains a more meaningful numerical representation, perhaps swap and move the current Fig 1A PC analysis to supp data??

It would be interesting to measure Ox phos effects to confirm whether predicted loss of function is observed. I suggest this as only some of the electron transport chain (ETC) components are described as altered. It is also worth noting that loss of ETC function may result in reduction in other functions despite energy production, for example lipid homeostasis, amino acid metabolism, Fe/S production and these could be considered within the data set. This would seem important as *C. albicans* do require ETC function for growth and has been linked to regulation of virulence traits.

Could the cytosine data suggest a mechanism to promote commensalism of *C. albicans* in the presence of actively growing *L. rhamnosus*? The switch from hyphae to yeast that is described in Fig 3D could suggest this, perhaps include a comment in the discussion around this idea?

We would like to thank the reviewers for their detailed and very constructive comments.

To deal with the diverse and comprehensive requests, we have performed a series of new experiments and analyses as listed below:

- 1. We performed dose-response experiments using lower concentrations of cytosine and analyzing its effect on both *C. albicans* filamentation and ability to cause damage to intestinal epithelial cells (IECs). We also analyzed the potential synergism with a blocked glycolysis or pentose phosphate pathway and with glucose depletion, as well as the effect of other nucleotides (adenine, guanine, thymine and uracil).**
- 2. The effect of *L. rhamnosus*-conditioned medium and 14 *L. rhamnosus*-derived metabolites on filamentation was assessed. The metabolites that had an impact on *C. albicans* filamentation were further analyzed in combination at lower concentrations for their effect on filamentation and *C. albicans* ability to damage IECs.**
- 3. We performed infection experiments with the *C. albicans mig1Δ* mutant. Additionally, the mutants *tye7Δ*, *gal4Δ* and *mig1Δ* were analyzed for their ability to cause damage to IECs with a repressed glycolysis pathway.**
- 4. The effect of *L. rhamnosus* on mitochondrial activity of *C. albicans* was assessed via Seahorse and XTT assays. Additionally, the potential of *L. rhamnosus*-derived metabolites to inhibit *C. albicans* mitochondrial activity was examined.**
- 5. We performed additional simulations accounting for metabolite uptake and secretion, biomass production, and sensitivity analysis.**

As a result, we provide additional data and further discussion as requested, which has certainly resulted in an improved manuscript. We further made a series of technical improvements, such as adding more information in the Methods section and figure legends, and full information on statistical tests. Finally, we have edited some sections for clarity and conciseness.

Our replies to the reviewer's comments are below, in bold:

Reviewer #1 (Remarks to the Author):

In this manuscript, Alonso-Roman and colleagues use "omics" to understand how the presence of a bacterium, *Lactobacillus rhamnosus*, reduces the ability of the fungus *Candida albicans* to cause epithelial cell damage.

It can be expected that bacteria and fungi antagonise each other by competing for nutrients and/or by secreting metabolites that impact on microbial growth, physiology, metabolism and/or development. Indeed, this study provides evidence for both of these possibilities. The novel aspects of the study include a comprehensive metabolomics and transcriptomics approach to profile a complex system consisting of all three relevant cell types (bacteria, fungi and host cells), showing that metabolites produced by host cells support bacterial growth enabling bacteria to protect them from *C. albicans*, the identification of cytosine's inhibitory effect on hyphal (but not yeast) growth of *C. albicans*, and the identification of several fungal mutants with reduced capacity to damage epithelial cells.

I appreciate that "omics" approaches will identify a complex picture, particularly in a biological scenario of co-culture of host, bacterial and fungal cells. As the authors conclude, this complexity might mean that a combination of several different mechanisms is involved in the inhibitory effects of *L. rhamnosus* on *C. albicans*. This study touches superficially on a few different mechanisms but does not explore any of the aspects in more depth. In my opinion, this is a weakness of the manuscript in its current form. Some additional in-depth analyses would be important to increase the value of the study beyond serving as a resource of the metabolic and transcriptional changes induced in *C. albicans* during co-culture with *L. rhamnosus* in the epithelial infection model.

Below, I am discussing some of the aspects that I found interesting and which could be explored more. Of course, not all of these questions can be addressed in one manuscript and I am not suggesting that they should be. But at least one of these aspects could be explored in more depth.

We thank the reviewer for the precise summary of our manuscript and for the thorough and constructive feedback provided.

(i) It is not surprising that the bacteria would deplete the nutrients needed for *C. albicans*, causing a metabolic shift in the fungus. As the authors suggest, this explains the reduced growth of *C. albicans* in the presence of *L. rhamnosus*. A key result in my view is shown in Figure 1, showing that the reduction of *C. albicans*-induced damage by *L. rhamnosus* goes beyond inhibition of fungal growth.

If I am not mistaken, it has been reported that *Lactobacillus* limits *C. albicans* hyphal formation.

The reviewer is correct, different authors have indeed reported that *Lactobacillus* sp. can limit *C. albicans* filamentation (for example MacAlpine *et al.* 2021 *Nat Commun*, Matsubara *et al.* 2016 *Appl Microbiol Biotechnol*, Allonsius *et al.* 2019 *Sci Rep*, Allonsius *et al.* 2017 *Microb Biotechnol*; see line 55 in the Introduction). This has been attributed to different factors (β -carboline, lactate, exopolysaccharides, chitin degradation). Similarly, we previously reported reduced hyphal length when *C. albicans* cells infected intestinal epithelial cells pre-colonized by *L. rhamnosus* (Graf *et al.* 2019 *Dis Mod Mech*).

To attribute the reduced filamentation to the effects of potential antivirulence metabolites produced by *L. rhamnosus*, we grew *C. albicans* in *L. rhamnosus*-conditioned supernatants. To exclude the role of the epithelium, and provide easier conditions for imaging, we grew *L. rhamnosus* in absence of epithelial cells using KBM supplemented with the key metabolites required to sustain *L. rhamnosus* growth (KBM+, described in Fig. 2; citric acid, carnitine and gamma-glutamyl alanine). When growing *C. albicans* in these conditioned supernatants, we observed impaired filamentation and a hyphae-to-yeast switch. These data have been included as Fig. S4A (lines 156-160) in the revised manuscript.

Representative images of *C. albicans* growth and morphology changes when grown in *L. rhamnosus*-conditioned (KBM/ *L.r.* or KBM+/ *L.r.*) or unconditioned media, after 20 h at 37°C with 5% CO₂ (n=3 biological replicates).

Here the authors show that cytosine (secreted by *L. rhamnosus*) is inhibitory for invasive hyphal growth (importantly, yeast-morphology cells could grow in the presence of cytosine). This is very interesting, but the mechanism by which cytosine specifically inhibits hyphal growth is unclear. While a full characterization of this mechanism is a whole new study and beyond the scope of here, some additional data would be beneficial. Are the concentrations of cytosine used in vitro (50mM) physiologically relevant?

Unfortunately, the answer to whether 50 mM cytosine is physiologically relevant is not simple. Cytosine production rates by *Lactobacillus* species or other bacteria have not been studied. However, we believe that the concentration used is higher than what one could expect in supernatants. However, it is even more difficult to estimate what concentrations could accumulate in close contact between bacteria and *C. albicans*, as we previously observed a close interaction between the fungus and *L. rhamnosus* (Graf *et al.* 2019 *Dis Mod Mech*). Nevertheless, *L. rhamnosus* in transwells separated from *C. albicans* still inhibited pathogenicity.

We have now also tested lower concentrations of cytosine, but we did not observe a reduction of damage at concentrations lower than 50 mM, while slighter effects on filamentation appear at 25 mM. To investigate the mechanism by which cytosine may reduce filamentation, we also analyzed other nucleotides. Only adenine exhibited some similar effect, inducing the formation of pseudohyphae rather than true hyphae. We also investigated the relationship of cytosine with metabolism by blocking the pentose phosphate pathway and using glucose-depleted media RPMI (supplemented with lactate) and adding the different nucleotides, but we found no additional effects. Additionally, we tested cytosine in KBM supplemented with 2-Deoxy-D-glucose in order to mimic the reduced flux in the glycolysis induced by *L. rhamnosus*, but these experimental conditions did not affect *C. albicans* responses to cytosine.

Have the authors tried any of the other metabolites secreted by *L. rhamnosus* for hyphal inhibition? In other words, is cytosine the key metabolite for hyphal inhibition?

The metabolites exclusively produced by *L. rhamnosus* during colonization of IEC are all candidates to inhibit *C. albicans* pathogenicity and potentially filamentation. In this aspect, cytosine was evaluated as one of them, yet several of the other metabolites in this cluster, have previously been described to inhibit filamentation and/or pathogenicity (lines 161-163 and 326 - 328).

From these we validated the metabolites alpha hydroxycaproate (HICA) and phenyllactate. While these metabolites potently affected hyphae formation (at lower concentrations than in published literature), these also substantially lowered the pH to levels at which *C. albicans* cells show reduced filamentation. Thus, the effect may have been indirect. Nevertheless, even when compared to pH-adjusted medium, reduced filamentation was still observed. A similar observation was made for indolelactate, which to our knowledge has not yet been described in the literature for its antifungal potential. We also found two additional metabolites that affected filamentation independent of pH changes: hydroxymethylbutyrate and phenylpyruvate. These data have been included as Fig. 3A.

This suggests that there is a combination of antivirulence metabolites in the microenvironment that *L. rhamnosus* generates, which could be driving the effects on filamentation that we describe. To validate this hypothesis, we assessed *C. albicans* filamentation in the presence of a combination of these six identified metabolites at lower concentrations (5 mM), plus lactate (15 mM, as previously measured in supernatants of colonized epithelial cells, Graf *et al.* 2019 *Dis Mod Mech*). We observed that the combination induced reduced filamentation and a mild switch towards yeast growth (Fig. 3B,3C S4). In addition, the combination of metabolites significantly reduced *C. albicans*-induced damage of IECs (Fig 3D).

With these data, we strongly believe that reduced filamentation and ability to cause damage is a multifactorial effect involving multiple changes induced in the environment by *L. rhamnosus*. (1) several antivirulence metabolites that impact *C. albicans* filamentous growth; (2) favored carbon and nitrogen sources are consumed and replaced by unfavored carbon and nitrogen sources. The Discussion has been adapted accordingly (lines 343-350).

(ii) The authors further suggest that the metabolic adaptation of *C. albicans* leads to differential expression of several genes that are required for causing epithelial cell damage (mutants analysed in Figure 6). This is interesting and re-enforces the idea that metabolism sits at the core of the adaptive programs that pathogens need to cause disease. However, the link between the metabolic reprogramming induced by *L. rhamnosus* and the differential expression of these genes remains unexplored. Transcriptomics identified several transcriptional and signalling regulators which could be explored further.

The authors selected the transcription factor Ace2 for some further analyses, showing that the *ace2* mutant causes reduced epithelial cell damage compared to the wild type strain. It is suggested here that Ace2 might be important for the observed metabolic regulation by *L. rhamnosus* because a study by Geraldine Butler's group (Mulhern et al 2006) showed that the *ace2* mutant shows some of the same metabolic changes as seen in the presence of *L. rhamnosus*. However, I am not sure if the effects of Ace2 on metabolism are direct.

Have the authors considered that the *ace2* mutant has a large cell morphology defect, which could be causative of the reduced host cell damage? In fact, Ace2 is best known as the activator of cell separation genes (the enzymes needed to degrade the cell wall following cell division). Therefore, the *ace2* mutant grows as cell chains and is likely to display cell wall changes that could impact invasion.

The differences in cell morphology between the wild type and *ace2* could also be driving some of the metabolic changes. The Butler lab paper also suggested that the metabolic genes differentially expressed in the *ace2* mutant are not likely to be direct targets of Ace2, due to lack of Ace2 binding sites.

The Discussion mentions differential expression of several other transcription factors with links to carbon metabolism (Mig1, Tye7 and Gal4) –these factors likely have more direct roles in metabolic regulation than Ace2. Have they been tested for epithelial cell damage or capacity to reprogram fungal metabolism in the presence of *L. rhamnosus*?

We thank the reviewer for raising this important point regarding the gene *ACE2*. To validate that the differently regulated genes during infection of epithelium colonized by *L. rhamnosus* could be intertwined with reduced pathogenicity, we screened corresponding deletion mutants from the Noble collection (Noble et al. 2010 *Nat Genetics*). We now have included a table of all screened mutants in the revised manuscript. In the screening, where these mutants were assessed for growth and damage capacity, the *ace2Δ* mutant was, among others, characterized as hypovirulent. We would like to clarify that this mutant screening did not serve as a basis for further analyses.

The reviewer is, however, correct that the *ace2Δ* mutant has a morphological defect that can explain the hypovirulence. The fact that the *ace2Δ* mutant also shows an altered metabolism (Mulhern et al. 2006 *Eukaryotic Cell*) is another hint that points to a direct

link between *C. albicans* pathogenicity and metabolism. The fact that genes important for filamentous growth, like *ACE2*, were down-regulated in the presence of *L. rhamnosus*, further strengthens the view that *L. rhamnosus* impacts filamentation.

We observed additional differential regulated genes that could explain the reduced filamentation and pathogenicity. For example, the repressor (of filamentous growth) genes *NRG1* and *TCC1* were significantly upregulated and the positive regulator (of filamentous growth) gene *SFL2* was downregulated early during infection. Furthermore, the gene *AHR1*, a regulator of pathogenicity factors such as the adhesin and invasin Als3 and Ece1 (Ruben *et al.* 2020 *mBio*), was downregulated. Nevertheless, other transcription factors positively regulating filamentation were up-regulated. We have adapted the Discussion to reflect the points raised by the reviewer and highlighted that the *ace2Δ* mutant has a filamentation defect.

Other mutants mentioned in the Discussion, such as *tye7Δ* and *gal4Δ*, were also present in our mutant screening (see Tab. S4 for all screened mutants). We adjusted Fig. 7A to indicate these two mutants in our screening. However, these deletion mutants did not show a significantly reduced virulence. We believe that individual deletions of these genes may be compensated by redundancy and metabolic flexibility of *C. albicans*. Supporting this, single KO mutants showed normal virulence, but a double mutant missing both *TYE7* and *GAL4* was hypovirulent in a published study (Askew *et al.* 2009 *PLoS Pathog*). A *mig1Δ* mutant was not included in the Noble collection, but based on the reviewer's comment, we tested its ability to induce damage in our model. Consistent with published data (Lagree *et al.* 2020 *PLoS Genetics*), a *mig1Δ* single mutant did not show reduced damage. Only when *MIG2* was deleted in addition to *MIG1* attenuated damage potential was observed.

We also tested mutants lacking these key transcription factors (Tye7, Gal4 and Mig1) for their ability to induce damage to epithelial cells in presence of 2-Deoxy-D-glucose, mimicking a suppressed glycolysis. However, this condition did not influence their ability to cause damage compared to the wild type (data shown here for the reviewer)

As the lack of one transcription factor seems to be by-passed without affecting *C. albicans* pathogenicity, we hypothesize that several metabolic transcription factors must be removed or downregulated in order to see a significant reduction in virulence (see Discussion, lines 374-378).

(iii) Transcriptome analyses suggested reduced mitochondrial respiration in the presence of *L. rhamnosus*. Lower respiratory capacity could reduce hyphal growth of *C. albicans* (this has been shown by multiple labs), but some experimental corroboration of the respiratory inhibition by *L. rhamnosus* would be required.

We thank the reviewer for stressing this interesting point. Unfortunately, it is not straightforward to assess *C. albicans* mitochondrial respiration in our model because it is difficult or impossible to separate fungal from the host cell mitochondrial respiration. In an experiment where we separated *C. albicans* from the host cells, Seahorse measurements did not show the characteristic curves of the mitostress test, highlighting

that further optimization is required to establish this methodology for *C. albicans* in our laboratory (data shown here for the reviewer). We propose that this goes beyond the scope of this revision.

To circumvent the problem, we used an XTT assay, which is primarily detecting the activity of mitochondrial enzymes. To avoid interference of bacterial and host cell metabolism, we used supernatants, conditioned media, and media supplemented with some of the metabolites produced by *L. rhamnosus*. As *L. rhamnosus*-conditioned medium showed an effect on *C. albicans* filamentation, we assessed mitochondrial activity at a very early timepoint. We observed that after one-hour exposure to *L. rhamnosus*-conditioned medium or supernatants, *C. albicans* mitochondrial activity was lower as compared to a corresponding control (exposure to unconditioned medium). To exclude effects on *C. albicans* growth, we normalized the mitochondrial activity to the biomass assessed by a crystal violet staining after the XTT assay. The mitochondrial activity normalized to biomass was also significantly lower when *C. albicans* cells had been grown in *L. rhamnosus*-conditioned medium. This new information has been now been included in the manuscript (lines 261-264, Fig. S7A, B).

Additionally, we screened metabolites produced when *L. rhamnosus* colonizes epithelial cells. We found two metabolites (uridine and 2-deoxyinosine) that did not affect filamentation or proliferation, but caused reduced mitochondrial activity compared to KBM only after 24h incubation. This information has also been included in the revised manuscript (lines 264-268, Fig. S7C).

(iv) The Results section on the transcriptomics data is in need of improvement. Neither Figure 4 nor Figure S5 go beyond high-level representation of the differentially expressed genes and GO enrichments. Further discussion and depiction of the specific metabolic genes that were affected,

as well as the regulators would be valuable for the reader. For example, transcriptional regulators of carbon metabolism are discussed in the Discussion (lines 319-326) but not mentioned in the Results.

As per suggestion, in the revised manuscript we extended our analysis of the transcriptional profiling. We discuss differential regulation of the most important transcription factor genes involving carbon (*GAL4*, *TYE7*, and *MIG1*) and amino-acid (*STP2*) metabolism. We also generated a heatmap to depict the differential regulation of glycolysis as well as the shift towards utilization of amino acids to feed the glyoxylate shunt and gluconeogenesis early during infection.

We agree with the reviewer that analysis of these specific pathways is more informative than the GO enrichments. We therefore included this information in the text of the Results section (lines 228-243) and added heatmaps as Fig. 5B and Fig. S7.

Reviewer #2 (Remarks to the Author):

The manuscript by Alonso-Roman et al investigates the interactions between *C. albicans*, *L. rhamnosus*, and intestinal epithelial cells. Using the transcriptomics, metabolomics, metabolic modelling and reverse genetics, revealed antivirulence and antifungal metabolic activity. Moreover, they observed epithelial cells promote the *L. rhamnosus* growth through specific metabolism and this showed affect the global metabolite profile available to the community and favoured *C. albicans* to metabolically reprogrammed which affect the virulence genes. I found the work timely and important considering the pathogenicity of *C. albicans*. The paper has generated in-vitro and in-silico data and integrate these data to support the findings. While I have found the results and the work timely and well presented, based on my acquired skills, I have few comments on the metabolic modelling section of the paper which I hope the authors find them constructive to improve their work, finding and making the analysis clearer.

We thank the reviewer for the constructive feedback and suggestions to improve the manuscript.

The authors mentioned they have done untargeted metabolomics however selected 235 metabolites. Could please elaborate more how these 235 metabolites were chosen as usually untargeted metabolomics produced more domain of metabolites. Also, how the metabolite profile was calculated considering the initial media? In what stage of the growth the sampling for the metabolites were done?

Indeed, untargeted metabolomics was performed. This was done with the commercial services offered by the company Metabolon. The library that Metabolon uses for spectral identification across several platforms contains thousands of metabolites, yet only metabolites for which there is a high confidence associated with their identification were included. The absence of specific metabolites from the dataset does not necessarily mean that those metabolites are not present, but indicates that they were below the detection limit of the assay. Furthermore, metabolite counts are influenced by the sample type itself and there is variability in the number of metabolites from study to study.

In our samples, 235 metabolites were identified by Metabolon after noise reduction and quality control. The values in the metabolome dataset are normalized raw area counts. Subsequently the data was rescaled for by setting the median of each metabolite at 1 and the values below the detection limit were imputed with the minimum. No absolute values of the metabolites were provided as this can only be done with targeted analysis where standards are also run in the analysis. This has been included in the Methods section of the revised manuscript.

Blank, incubated, culture medium was included in the analysis to get an insight into the metabolites that are present in the culture medium, and how these are influenced by the incubation under cell-culture conditions. At each time point the scaled metabolite data were used to generate heatmaps, which were subsequently analyzed by unsupervised hierarchical clustering using the complete linkage agglomeration method. Because we analyzed the supernatant metabolite profiles of each of the players individually, in combinations, as well as blank cell culture medium, we were able to make predictions,

about which metabolites were produced or consumed by the individual players or combinations. A similar analysis was provided by Metabolon, which, to most extent, overlapped with our clustering analysis. However, as we did not know the underlying calculation methods that were used for this analysis we decided to use our the clustering analysis.

The sampling of the metabolites was performed at 6 and 12 hours post infection. This has now been indicated in the Results section. These time points were selected because they are at crucial stages in the infection process. After six hours, *C. albicans* has initiated filamentous growth and invades the intestinal epithelial cells, but no extensive tissue damage is induced (Allert *et al.* 2018 *mBio*). Whereas at 12 hours post infection, the fungus starts to damage intestinal epithelial cells with its secreted peptide toxin candidalysin (Allert *et al.* 2018 *mBio*). However, in the conditions where we investigate *Lactobacillus*-colonized epithelium the epithelial cells were colonized 18 hours prior to infection. We previously characterized that this prior inoculation is required to establish a stable bacterial colonization on the epithelial cells, otherwise significantly higher bacterial inocula are required for the protective effect (Graf *et al.* 2019 *Dis Mod Mech*).

-For the GEMs and metabolic modelling, the authors have referred to the already generated models and cites two papers. However, looking at VMH and also the RECON3D paper, there are no specific *C. albicans* and Epithelial models. I guess the human model has been used for the intestinal epithelial model. This needs to be clearly addressed that the human generic model has been used otherwise there are other available tissue specific model that can be applied here such as <https://metabolicatlas.org/gems/repository>

Indeed, we referred to three GEMs in the methods section. We apologize if the references to the used GEMs were misleading. We used the *C. albicans* GEM from our recent publication (Mirhakkak *et al.* 2020 *ISME J*), which is not available at www.vmh.life, but can be downloaded from the supplementary materials of the manuscript. We specified the description in the Methods section as follows (line 550): "Specifically, the recently published model for *C. albicans*³⁷ was downloaded from the supplementary material of the publication. The GEMs for *Lactobacillus rhamnosus* LMS2-1 for *L. rhamnosus*³¹, and Recon3D 3.01³⁰, a comprehensive generic GEM of human metabolism used to simulate human intestinal epithelial cells, were downloaded from www.vmh.life."

-The simulation performed by each of the organisms, the constraints and objective functions have not specified in the paper and just mentioned in the methods "In brief, feasible uptake rates for available metabolites were adapted from the metabolome measurements across all investigated conditions." Please support this analysis with metabolite uptake and secretion and this analysis need to be supported as well with the microbial biomass productions and sensitivity analysis.

We reformulated our description to clarify our approach for our simulations and added biomass production and sensitivity analysis results to the supplements. Specifically, we clarified how we used FBA and FVA in conjunction with our metabolomics data. The respective Methods section now reads (line 557): "Feasible uptake flux ranges for each metabolite in our GEMs were kept in the range [0, 1000] mmol/g(DW)h. The metabolite concentrations for each sample were transformed into this range based on the metabolite glutamine showing the highest concentration in the 12 hpi *L. rhamnosus* supernatant compared to all measured metabolites and all samples. The uptake rate of glutamine was set to 1000 mmol/g(DW)h accordingly, whereas all others were set to the respective proportion to the maximum glutamine value. The biomass function of each GEM was used as objective function for all metabolic modelling simulations. To obtain objective function values mimicking an anaerobic environment (oxygen influx prohibited) as well as feasible reaction flux ranges supporting at least 90% of the objective function flux, we applied flux balance analysis (FBA) and flux variability analysis (FVA) across all tested conditions for all tested GEMs^{29,82}."

In addition, we added remarks for using FBA or FVA as well as more references to the Methods section throughout the results of our manuscript.

We created an additional supplementary data file with biomass objective function values and associated flux ranges for all reactions for all simulations and investigated media conditions (supplementary Data 2).

Towards sensitivity analysis we agree with the reviewer that this would increase trust in our analysis. Towards this point, we created another supplementary material file where we included two simulation sets over different fractions of required objective function values (supplementary Data 3). 1) a sensitivity analysis for the fraction of required objective function flux when analyzing secretion and uptake capabilities of IEC and *L. rhamnosus* GEMs (related to Fig. 3E and S4). 2) pathway activity changes when requiring different cut-offs for flux differences. For 1) we saw no and for 2) only minor effects on the results. Given these insights we concluded our results are robust in the vicinity of the chosen optimization fractions of 10% shown in the manuscript.

Of note, along creating these supplementary materials, we realized that sink reactions present in the Recon3D 3.01 model were allowed to carry flux by default. As these reactions are present mainly for debugging issues, we blocked the erroneous influx of some metabolites by these reactions accordingly and reran our simulations. Although these simulations resulted in a more notable shift in pathway activity for IECs upon different simulated media compared to our initial submission, it did not change the theme of our work, as IECs kept showing notable less pathway activity change compared to *L. rhamnosus* when simulated on the same media (Fig. 2G and S3). We updated Fig. S3A accordingly and apology that we oversaw this artefact in our initial submission. We added the following passage to our results to describe the few pathway activity changes in IECs we do see: "In comparison, changes in metabolic pathway activity were less present in IECs (Fig. S3A). Here, only a few amino acids (including tyrosine and phenylalanine), ubiquinone and taurine pathways next to the generic protein assembly/degradation metabolic subsystem showed shifts in pathway activity of at least 40% upon *L. rhamnosus* affected supernatants. These simulation results suggest that *L. rhamnosus* utilizes IEC-secreted metabolites without triggering metabolic changes in the host to the same extent it undergoes itself. These results did not change upon varying thresholds for required flux activity changes in the compared conditions (supplementary Data 3)." Fig. 3G and S4 were also not affected by resimulating IECs on different supernatant media. As part of our revision efforts we updated Fig. S5, which now is consistent with shown comparisons in the related Fig. 3G.

-How the conversion of the untargeted metabolites without unit of concentration to a flux unit for input has been done? Can you support this calculation and what is the assumption for the dilution rate? In the method it also mentioned the modelling were done anaerobically. Was this the case for the epithelial and Candida model?

We apologize if the description was not clear on how we converted the given concentration differences per sample to flux uptake rates. For the sake of model simplification and since metabolic model analysis is based on the steady state assumption of all fluxes (meaning the model simulation is not allowed to deplete or accumulate any metabolite except for metabolites taken up or secreted *via* exchange reactions) we assumed the likelihood of uptake correlates with the measured metabolite concentrations, while we neglected dilution rates. We successfully used this approach before (Weis *et al.* 2017 Cell), as it allows a compromise between given metabolomics concentrations and feasible model simulations, when only differences in feasible flux, and thus pathway, activities are of interest. All model simulations for all GEMs were done mimicking anaerobic growth and clarified this in the Methods description (see modified passage in the comment above).

-Has the author tried to perform pairwise or community modelling to simulate the interactions between the three organisms at once? This type of modelling can be more suitable for this study. There are already available functions in the COBRA to perform these analyses.

We thank the reviewer for this remark. We indeed simulate paired models and investigated uptake and secretion capabilities of our GEMs compared to the single GEM simulation results shown in Fig. 3G and S4. We did not observe any difference in the secretion or uptake capabilities. We furthermore tested a community modeling approach, which suffered from non-trivial EGC cycles across multiple species and

resulted in unrealistic flux predictions. Given the complex nature of simulating the correct (or a variable) number of cells for each of our three investigated GEMs (according to our investigated species) towards a combined community biomass function with weights for each organism and due to the compatible *in silico* simulations we obtained following a more straightforward approach with investigating one model simulation at a time, we opted against showing community modeling simulations in our manuscript. We, nevertheless, agree that this is an interesting avenue for follow-up studies and mentioned this in our discussion accordingly (line 356): "Although we followed a one-model-at-a-time simulation approach, our *in silico* analysis revealed changes in key metabolic pathways and regulator genes of *C. albicans*, which we also found in our *C. albicans* transcriptome data. Further more sophisticated community modelling simulating all three GEMs simultaneously were largely in agreement with our single GEM simulations, but may be investigated in more depth in future work."

-Based on the method, the pathway enrichment analysis was done using Revigo. What type of statistics were performed to report the significant ones?

From the reviewer question it became apparent to us that the Methods section on this needed revision. The GO-term enrichment was analyzed using the GO-Term Finder on Candida genome database (Skrzypek *et al.* 2017 Nucl Acid Res), which uses a hypergeometric distribution with Multiple Hypothesis Correction (Bonferroni Correction) to calculate *p*-values. Subsequently, the significantly enriched GO terms were processed using REVIGO (Supek *et al.* 2011 PLoS One; settings: similarity: Tiny (0.4); database: whole Uniprot; semantic similarity measure: SimRel) to remove overlapping and redundant GO-terms. We have adapted this in the methods section accordingly.

Reviewer #3 (Remarks to the Author):

The authors are to be congratulated on what is an elegant study that advances our knowledge of host/microbe interactions as well as providing what I see as a template for future studies of this nature. The paper is very well written and the data are analysed in a manner that fully supports the statements made within the manuscript. The data are of a high quality and the rationale for experimental design is very clear.

Although I believe the paper could be accepted as it is now I would make some small suggestions as points for consideration.

We would like to thank the reviewer for the constructive comments and the feedback to improve the manuscript.

* I feel it would be better to include FigS1A in place of Figure 1A as it contains a more meaningful numerical representation, perhaps swap and move the current Fig 1A PC analysis to supp data??

We assume that the reviewer means Fig. 2A instead of the Fig. 1A. We appreciate the reviewer's suggestion, but we believe that a PCA plot reflects better the dynamics in the experiment and variability among the different conditions and experimental replicates, a dimension which is lost in the Venn Diagrams. As we appreciate the that the numerical representation is also informative, we moved these data into the main figure as Fig. 2B.

* It would be interesting to measure Ox phos effects to confirm whether predicted loss of function is observed. I suggest this as only some of the electron transport chain (ETC) components are described as altered.

We thank the reviewer for this interesting suggestion. Exactly this point has also been raised by reviewer 1, please see the detailed response above. In brief, OXPHOS measurements using Seahorse are not straightforward and require further extensive optimization for *C. albicans*. Therefore, we used XTT assays which assess the activity of the mitochondrial enzymes. We observed that *L. rhamnosus*-conditioned supernatants as well as selected metabolites reduced activity of the mitochondrial enzymes assessed by the XTT assay (lines 261-267, Fig. S7A, B, C). The transcriptional profiling revealed that complex 1 and 4 of the ETC show an overall trend to be regulated although the

individual genes did not reach significance, to illustrate this we generated a heatmap showing expression of the individual complexes of the ETC (Fig. S7D).

It is also worth noting that loss of ETC function may result in reduction in other functions despite energy production, for example lipid homeostasis, amino acid metabolism, Fe/S production and these could be considered within the data set. This would seem important as *C. albicans* do require ETC function for growth and has been linked to regulation of virulence traits.

We specifically looked at genes in the Iron-sulfur cluster and found that the genes for the two predicted mitochondrial matrix localized proteins orf19.1267.1 and *SSQ1* were significantly downregulated at 6 hpi. In contrast *ISU1*, which is predicted to play a role in the iron sulfur-cluster was upregulated. Regarding lipid homeostasis the gene *PEL1*, predicted to be involved in mitochondrial phospholipid biosynthesis, was significantly downregulated.

While these are definitely interesting genes that strengthen the idea that mitochondrial function is impaired, we feel that reporting on these genes in our manuscript, which already has several messages, may overload the reader. We, therefore, decided to keep the analysis with only the genes of the mitochondrial complexes now included as Fig. S7D.

* Could the cytosine data suggest a mechanism to promote commensalism of *C. albicans* in the presence of actively growing *L. rhamnosus*? The switch from hyphae to yeast that is described in Fig 3D could suggest this, perhaps include a comment in the discussion around this idea?

The reviewer addresses the key point of our manuscript. Commensalism is a phenotype that is extremely difficult to model using *in vitro* systems where *C. albicans* by default behaves extremely pathogenic. Using actively growing *L. rhamnosus* cells on intestinal epithelial cells we established a situation mimicking commensalism.

Indeed, metabolites such as cytosine, which inhibit *C. albicans* pathogenicity mechanisms may be key in promoting commensalism. Nevertheless, as also mentioned in the response to reviewer 1, we would like to stress that we believe that promotion of commensalism is likely a multifactorial process and can possibly not be attributable to a single metabolite. As we discussed, a variety of the metabolites that we observed upon colonization of intestinal epithelial cells with *L. rhamnosus*, have previously been described to antagonize *C. albicans* pathogenicity. In addition, the revised manuscript now describes several additional metabolites that impact filamentation or mitochondrial activity.

We have included comments in our Discussion to stress that interactions between bacteria and *C. albicans* that induce a transition from hyphae to yeast may likely play key roles in the commensalism promoting feature of a healthy microbiome (lines 343-350).

REFERENCES

Allert S, Förster TM, Svensson CM, Richardson JP, Pawlik T, Hebecker B, Rudolphi S, Juraschitz M, Schaller M, Blagojevic M, Morschhäuser J, Figge MT, Jacobsen ID, Naglik JR, Kasper L, Mogavero S, Hube B. Candida albicans-Induced Epithelial Damage Mediates Translocation through Intestinal Barriers. mBio. 2018 Jun 5;9(3):e00915-18. doi: 10.1128/mBio.00915-18. PMID: 29871918; PMCID: PMC5989070.

Allonsius CN, van den Broek MFL, De Boeck I, Kiekens S, Oerlemans EFM, Kiekens F, Foubert K, Vandenheuvel D, Cos P, Delputte P, Lebeer S. Interplay between Lactobacillus rhamnosus GG and Candida and the involvement of exopolysaccharides. Microb Biotechnol. 2017 Nov;10(6):1753-1763. doi: 10.1111/1751-7915.12799. Epub 2017 Aug 3. PMID: 28772020; PMCID: PMC5658588.

Allonsius CN, Vandenheuvel D, Oerlemans EFM, Petrova MI, Donders GGG, Cos P, Delputte P, Lebeer S. Inhibition of *Candida albicans* morphogenesis by chitinase from *Lactobacillus rhamnosus* GG. *Sci Rep*. 2019 Feb 27;9(1):2900. doi: 10.1038/s41598-019-39625-0. PMID: 30814593; PMCID: PMC6393446.

Askew C, Sellam A, Epp E, Hogues H, Mullick A, Nantel A, Whiteway M. Transcriptional regulation of carbohydrate metabolism in the human pathogen *Candida albicans*. *PLoS Pathog*. 2009 Oct;5(10):e1000612. doi: 10.1371/journal.ppat.1000612. Epub 2009 Oct 9. PMID: 19816560; PMCID: PMC2749448.

Graf K, Last A, Gratz R, Allert S, Linde S, Westermann M, Gröger M, Mosig AS, Gresnigt MS, Hube B. Keeping *Candida* commensal: how lactobacilli antagonize pathogenicity of *Candida albicans* in an *in vitro* gut model. *Dis Model Mech*. 2019 Sep 12;12(9):dmm039719. doi: 10.1242/dmm.039719. PMID: 31413153; PMCID: PMC6765188.

Lagree K, Woolford CA, Huang MY, May G, McManus CJ, Solis NV, Filler SG, Mitchell AP. Roles of *Candida albicans* Mig1 and Mig2 in glucose repression, pathogenicity traits, and SNF1 essentiality. *PLoS Genet*. 2020 Jan 21;16(1):e1008582. doi: 10.1371/journal.pgen.1008582. PMID: 31961865; PMCID: PMC6994163.

MacAlpine J, Daniel-Ivad M, Liu Z, Yano J, Revie NM, Todd RT, Stogios PJ, Sanchez H, O'Meara TR, Tompkins TA, Savchenko A, Selmecki A, Veri AO, Andes DR, Fidel PL Jr, Robbins N, Nodwell J, Whitesell L, Cowen LE. A small molecule produced by *Lactobacillus* species blocks *Candida albicans* filamentation by inhibiting a DYRK1-family kinase. *Nat Commun*. 2021 Oct 22;12(1):6151. doi: 10.1038/s41467-021-26390-w. PMID: 34686660; PMCID: PMC8536679.

Matsubara VH, Wang Y, Bandara HMHN, Mayer MPA, Samaranyake LP. Probiotic lactobacilli inhibit early stages of *Candida albicans* biofilm development by reducing their growth, cell adhesion, and filamentation. *Appl Microbiol Biotechnol*. 2016 Jul;100(14):6415-6426. doi: 10.1007/s00253-016-7527-3. Epub 2016 Apr 18. PMID: 27087525.

Mirhakkak MH, Schäuble S, Klassert TE, Brunke S, Brandt P, Loos D, Uribe RV, Senne de Oliveira Lino F, Ni Y, Vylkova S, Slevogt H, Hube B, Weiss GJ, Sommer MOA, Panagiotou G. Metabolic modeling predicts specific gut bacteria as key determinants for *Candida albicans* colonization levels. *ISME J*. 2021 May;15(5):1257-1270. doi: 10.1038/s41396-020-00848-z. Epub 2020 Dec 15. PMID: 33323978; PMCID: PMC8115155.

Mulhern SM, Logue ME, Butler G. *Candida albicans* transcription factor Ace2 regulates metabolism and is required for filamentation in hypoxic conditions. *Eukaryot Cell*. 2006 Dec;5(12):2001-13. doi: 10.1128/EC.00155-06. Epub 2006 Sep 22. PMID: 16998073; PMCID: PMC1694816.

Noble SM, French S, Kohn LA, Chen V, Johnson AD. Systematic screens of a *Candida albicans* homozygous deletion library decouple morphogenetic switching and pathogenicity. *Nat Genet*. 2010 Jul;42(7):590-8. doi: 10.1038/ng.605. Epub 2010 Jun 13. PMID: 20543849; PMCID: PMC2893244.

Ruben S, Garbe E, Mogavero S, Albrecht-Eckardt D, Hellwig D, Häder A, Krüger T, Gerth K, Jacobsen ID, Elshafee O, Brunke S, Hünninger K, Kniemeyer O, Brakhage AA, Morschhäuser J, Hube B, Vylkova S, Kurzai O, Martin R. Ahr1 and Tup1 Contribute to the Transcriptional Control of Virulence-Associated Genes in *Candida albicans*. *mBio*. 2020 Apr 28;11(2):e00206-20. doi: 10.1128/mBio.00206-20. PMID: 32345638; PMCID: PMC7188989.

Skrzypek MS, Binkley J, Binkley G, Miyasato SR, Simison M, Sherlock G. The *Candida* Genome Database (CGD): incorporation of Assembly 22, systematic identifiers and visualization of high throughput sequencing data. *Nucleic Acids Res*. 2017 Jan 4;45(D1):D592-D596. doi: 10.1093/nar/gkw924. Epub 2016 Oct 13. PMID: 27738138; PMCID: PMC5210628.

Supek F, Bošnjak M, Škunca N, Šmuc T. REVIGO summarizes and visualizes long lists of gene ontology terms. *PLoS One*. 2011;6(7):e21800. doi: 10.1371/journal.pone.0021800. Epub 2011 Jul 18. PMID: 21789182; PMCID: PMC3138752.

Weis S, Carlos AR, Moita MR, Singh S, Blankenhaus B, Cardoso S, Larsen R, Rebelo S, Schäuble S, Del Barrio L, Mithieux G, Rajas F, Lindig S, Bauer M, Soares MP. Metabolic Adaptation Establishes Disease Tolerance to Sepsis. *Cell*. 2017 Jun 15;169(7):1263-1275.e14. doi: 10.1016/j.cell.2017.05.031. PMID: 28622511; PMCID: PMC5480394.

Reviewers' Comments:

Reviewer #1:

Remarks to the Author:

The revised manuscript addressed my suggestions in a satisfactory manner. I have no further issues.

Reviewer #2:

Remarks to the Author:

Thanks for your extensive response to my comments and adding more analysis. The method section now contains the necessary information for following the metabolic modelling work in the paper.

Reviewer #3:

Remarks to the Author:

The reviewers comments have been addressed thoroughly with a combination of considered rebuttal, experiments and text changes, my recommendation would be to accept the paper